# Towards a Unified Analysis of Kernel-based Methods Under Covariate Shift

**Xingdong Feng**[1], **Xin He**[1], **Caixing Wang**[1][*] **Chao Wang**[1], **Jingnan Zhang**[2]
[1]School of Statistics and Management, Shanghai University of Finance and Economics
[2]International Institute of Finance, School of Management,
University of Science and Technology of China
`{feng.xingdong, he.xin17}@mail.shufe.edu.cn`
`{wang.caixing, wang.chao}@stu.shufe.edu.cn`
`jnzhang@ustc.edu.cn`

## Abstract

Covariate shift occurs prevalently in practice, where the input distributions of the source and target data are substantially different. Despite its practical importance in various learning problems, most of the existing methods only focus on some specific learning tasks and are not well validated theoretically and numerically. To tackle this problem, we propose a unified analysis of general nonparametric methods in a reproducing kernel Hilbert space (RKHS) under covariate shift. Our theoretical results are established for a general loss belonging to a rich loss function family, which includes many commonly used methods as special cases, such as mean regression, quantile regression, likelihood-based classification, and margin-based classification. Two types of covariate shift problems are the focus of this paper and the sharp convergence rates are established for a general loss function to provide a unified theoretical analysis, which concurs with the optimal results in literature where the squared loss is used. Extensive numerical studies on synthetic and real examples confirm our theoretical findings and further illustrate the effectiveness of our proposed method.

## 1 Introduction

Covariate shift is a phenomenon that commonly occurs in machine learning, where the distribution of input features (covariates) changes between the source (or training) and target (or test) data, while the conditional distribution of output values given covariates remains unchanged (Shimodaira, 2000; Pan & Yang, 2010). Such a phenomenon is illustrated in Figure 1 where the learned predictive function from the source data may significantly differ from the true function. Thus, the prediction performance can be largely degraded since the predictive function has not been trained on data that accurately represents the target environment. Covariate shift can arise in a variety of domains, including medicine and healthcare (Wei et al., 2015; Hajiramezanali et al., 2018), remote sensing (Tuia et al., 2011), and natural language and speech processing (Yamada et al., 2009; Fei & Liu, 2015). Various factors contribute to covariate shift, such as data sampling bias (e.g., patient demographics in healthcare applications), changes in the studied population (e.g., vocabulary evolution in natural language processing), or measurement and observational errors (e.g., sensor noise or calibration errors in the sensor domain). Compared to the well-studied supervised learning without such a distribution mismatch (Györfi et al., 2002), there still exists some gap in both theoretically and numerically understanding the influence of the covariate shift under various kernel-based learning problems.

---

[*]Caixing Wang is the corresponding author and all the authors contributed equally to this paper and their names are listed in alphabetical ordering.

37th Conference on Neural Information Processing Systems (NeurIPS 2023).

This paper provides a unified analysis of the kernel-based methods under covariate shift. Moreover, a general loss function is considered, which is allowed to belong to a rich loss function family and thus includes many commonly used methods as special cases, such as mean regression, quantile regression, likelihood-based classification, and margin-based classification. This paper considers two types of covariate shift problems in which the importance ratio is assumed to be uniformly bounded or to have a bounded second moment. A unified theoretical analysis has been provided that the sharp convergence rates are established for a general loss function under two evaluation metrics, which concurs with the optimal results in Ma et al. (2023) where the squared loss is used. Our theoretical findings are also validated by a variety of synthetic and real-life examples.

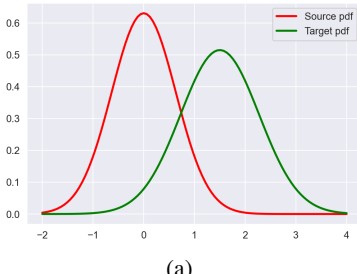 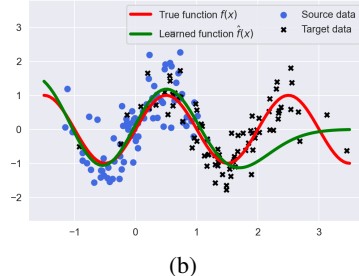

(a)                                           (b)

Figure 1: (a) The probability density functions of normal distributions with $\mu_1 = 0, \sigma^2 = 0.4$ that the source data is driven from and $\mu_1 = 1.5, \sigma^2 = 0.6$ that the target data is driven from, respectively; (b) the learned function trained by using the source data and the true mean regression function. Note that the considered example serves as an illustration that satisfies case (ii) in Section 2.3.

**Our contributions.** The contributions of this paper are multi-fold. We propose a unified analysis of the kernel-based methods under covariate shift, which provides an insightful understanding of the influences of covariate shift on the kernel-based methods both theoretically and numerically. By utilizing empirical process in learning theory, we investigate the theoretical behaviors of several estimators under various conditions, with particular emphasis on different importance ratio cases between the source and target distributions. Specifically, we show that the unweighted estimator achieves the optimal learning rates in terms of both $\mathcal{L}^2$-error and the excess risk with respect to the target distribution under the uniformly bounded case. Yet, the unweighted estimator is sub-optimal under the bounded second moment case. Then, we construct a weighted estimator by using an appropriate truncated ratio, which again attains a sharp convergence rate. Unlike kernel ridge regression (KRR), the estimator for the general loss does not have an explicit form, making many classical analysis techniques inapplicable (Lian, 2022) and in our technical proofs, the theoretical derivation is much more involved due to the considered general loss function. Numerous experiments on synthetic data and multi-source real data with various loss functions confirm our theoretical findings. To some extent, this paper provides a comprehensive study of the numerical performance of the kernel-based methods under various scenarios with covariate shift.

**Related work.** Some most related works including covariate shift adaptation and importance ratio estimation are presented below.

**Covariate shift adaptation.** Shimodaira (2000) investigates the impact of covariate shift in parametric regression with maximum likelihood estimation and proposes an importance-weighting (IW) estimator, which has a significant improvement when the misspecification does exist. Sugiyama & Müller (2005b) further extend this work by analyzing an unbiased estimator for $\mathcal{L}^2$-error. These fundamental works also motivate a variety of follow-up studies under the parametric setting Sugiyama & Storkey (2006); Yamazaki et al. (2007); Wen et al. (2014); Lei et al. (2021). Beyond the parametric setting, Kpotufe & Martinet (2021) consider the nonparametric classification problem and provide a new minimax result that concisely captures the relative benefits of source and target labeled data under covariate shift. Focusing on the Hölder continuous regression function class, Pathak et al. (2022) propose a new measure of distribution mismatch between the source and target distributions. Ma et al. (2023) and Gogolashvili et al. (2023) establish the optimal learning rates for KRR over RKHS. Recently, covariate shift in overparameterized models, such as high-dimensional models and neural networks, has also drawn tremendous attention from relevant researchers (Hendrycks & Dietterich, 2019; Byrd & Lipton, 2019; Hendrycks et al., 2021; Tripuraneni et al., 2021).

**Importance ratio estimation.** A straightforward approach to estimating the importance ratio is separately estimating the source and target densities by kernel density estimation (Sugiyama & Müller, 2005a; Baktashmotlagh et al., 2014) and then computing the ratio. In practice, it is more efficient to directly estimate the importance ratio by minimizing some discrepancy measures between distributions, including kernel mean matching (Huang et al., 2006; Gretton et al., 2009), Kullback-Leibler divergence (Sugiyama et al., 2007a,b, 2012) and non-negative Bregman divergence (Kato & Teshima, 2021).

**Notation.** In this paper, we use $C$ to denote a universal constant that may vary from line to line. For two sequence $a_n$ and $b_n$, we say $a_n \asymp b_n$ if and only if $a_n = O(b_n)$ and $b_n = O(a_n)$ hold.

## 2 Problem formulation

### 2.1 Learning within a rich family of loss functions

Suppose that the random pair $(\mathbf{x}, y)$ is drawn from some unknown distribution $P_{\mathbf{x},y}$ with $\mathbf{x} = (x_1, \ldots, x_p)^\top \in \mathcal{X}$, where $\mathcal{X} \subset \mathcal{R}^p$ is a compact support and $y \in \mathcal{Y} \subset \mathcal{R}$. In the literature of machine learning, the true target function $f^*$ is often defined as the minimizer of the expected risk with a pre-specified loss function $L(\cdot, \cdot) : \mathcal{Y} \times \mathcal{R} \to \mathcal{R}^+$ that

$$f^* := \operatorname{argmin}_f \mathcal{E}^L(f) = \operatorname{argmin}_f E\left[L\big(y, f(\mathbf{x})\big)\right]. \tag{1}$$

Throughout this paper, we consider a general loss function belonging to a rich loss function family that $L(y, \cdot)$ is assumed to be convex and locally $c_L$-Lipschitz continuous (Wainwright, 2019; Dasgupta et al., 2019), that is for some $V \geq 0$, there exists a constant $c_L > 0$ such that $|L(y, \omega) - L(y, \omega')| \leq c_L|\omega - \omega'|$ holds for all pairs $\omega, \omega' \in [-V, V]$ and $y \in \mathcal{Y}$. A variety of popularly used loss functions satisfy these two conditions, including

- **Squared loss**: $L(y, f(\mathbf{x})) = (y - f(\mathbf{x}))^2$ with $c_L = 2(M_y + V)$, for any $|y| \leq M_y$ with constant $M_y > 0$;
- **Check loss**: $L(y, f(\mathbf{x})) = (y - f(\mathbf{x}))(\tau - I_{\{y \leq f(\mathbf{x})\}})$ with $c_L = 1$ and quantile level $\tau$;
- **Huber loss**: $L(y, f(\mathbf{x})) = (y - f(\mathbf{x}))^2$, if $|y - f(\mathbf{x})| \leq \delta; \delta|y - f(\mathbf{x})| - \frac{1}{2}\delta^2$, otherwise, with $c_L = \delta$;
- **Logistic loss**: $L(y, f(\mathbf{x})) = \log(1 + \exp(-yf(\mathbf{x})))/\log 2$ with $c_L = (\log 2)^{-1}e^V/(1 + e^V)$;
- **Hinge loss**: $L(y, f(\mathbf{x})) = (1 - yf(\mathbf{x}))_+$ with $c_L = 1$.

Note that the explicit form of $f^*$ may differ from one loss to another. For example, for the squared loss, $f^*(\mathbf{x}) = E[y|\mathbf{x}]$; for the check loss, $f^*(\mathbf{x}) = Q_\tau(y|\mathbf{x})$ with $Q_\tau(y|\mathbf{x}) = \inf\{y : P(Y \leq y|\mathbf{x}) \geq \tau\}$; for the hinge loss, $f^* = \operatorname{sign}(P(y = 1|\mathbf{x}) - 1/2)$ with $\operatorname{sign}(\cdot)$ denoting the sign function. Moreover, we require $f^* \in \mathcal{H}_K$, where $\mathcal{H}_K$ denotes the RKHS induced by a pre-specified kernel function $K(\cdot, \cdot) : \mathcal{X} \times \mathcal{X} \to \mathcal{R}$. In practice, one of the most important tasks in machine learning is to learn $f^*$ from the source data and use the estimated predictive function $\widehat{f}$ for prediction in the target data.

### 2.2 Measurement under source and target distributions

Classical learning problems often explicitly or implicitly assume that the source and target data are drawn from the same distribution. Precisely, the source data comes from a joint distribution $P_S = P_{y|\mathbf{x}}P_{\mathbf{x}}^S$ where $P_{y|\mathbf{x}}$ is the conditional distribution and $P_{\mathbf{x}}^S$ is the input distribution with density function $\rho_{\mathbf{x}}^S$. Then, the performance of the estimated predictive function $\widehat{f}$ is usually evaluated by the $\mathcal{L}^2(P_{\mathbf{x}}^S)$-error or the excess risk error with respect to $P_S$ that

$$\|\widehat{f} - f^*\|_S^2 = E_{\mathbf{x} \sim S}\left[\big(\widehat{f}(\mathbf{x}) - f^*(\mathbf{x})\big)^2\right], \quad \mathcal{E}_S^L(\widehat{f}) - \mathcal{E}_S^L(f^*) = E_S\left[L\big(y, \widehat{f}(\mathbf{x})\big) - L\big(y, f^*(\mathbf{x})\big)\right],$$

where $E_{\mathbf{x} \sim S}$ and $E_S$ are the expectation over $P_{\mathbf{x}}^S$ and $P_S$ conditioning on the observed data, respectively. When covariate shift occurs, the target data may come from some totally different joint

distribution $P_T = P_{y|\mathbf{x}} P_{\mathbf{x}}^T$ in the sense that although $P_{y|\mathbf{x}}$ representing the regression or classification rule remains invariant, $P_{\mathbf{x}}^T$ differs significantly from $P_{\mathbf{x}}^S$. We also assume that $P_{\mathbf{x}}^T$ has density function $\rho_{\mathbf{x}}^T$. Obviously, it is reasonable to evaluate $\widehat{f}$ under $P_T$ instead of $P_S$. Then, our primary interest is the $\mathcal{L}^2(P_{\mathbf{x}}^T)$-error or the excess risk error with respect to $P_T$ that

$$\|\widehat{f} - f^*\|_T^2 = E_{\mathbf{x}\sim T}\big[\big(\widehat{f}(\mathbf{x}) - f^*(\mathbf{x})\big)^2\big], \quad \mathcal{E}_T^L(\widehat{f}) - \mathcal{E}_T^L(f^*) = E_T\big[L(y, \widehat{f}(\mathbf{x})) - L(y, f^*(\mathbf{x}))\big],$$

where $E_{\mathbf{x}\sim T}$ and $E_T$ are the expectation over $P_{\mathbf{x}}^T$ and $P_T$ conditioning on the observed data, respectively. In the rest of this paper, $f^*$ is defined as the optimal function under the target distribution such that $f^* = \operatorname{argmin}_f E_T[L(y, f(\mathbf{x}))]$.

## 2.3 Kernel-based estimation under covariate shift and importance ratio correction

Suppose that random sample $\mathcal{Z}_n^S = \big\{\big(\mathbf{x}_i^S, y_i^S\big)\big\}_{i=1}^n$ are i.i.d. drawn from the source distribution $P_S$. We consider the classical nonparametric estimation problem in RKHS (Vapnik, 1999) that

$$\widehat{f} := \operatorname*{argmin}_{f\in\mathcal{H}_K} \frac{1}{n}\sum_{i=1}^n L\big(y_i^S, f(\mathbf{x}_i^S)\big) + \lambda\|f\|_K^2, \tag{2}$$

where $\lambda$ is the regularization parameter. Without covariate shift, the first term on the right side of (2) is an unbiased estimator of $\mathcal{E}_T^L(f)$. However, it becomes biased when covariate shift occurs, and thus may lead to inaccurate predictive estimator. To tackle this issue, we consider the importance ratio measuring the discrepancy between distributions that is $\phi(\mathbf{x}) = \rho_{\mathbf{x}}^T(\mathbf{x})/\rho_{\mathbf{x}}^S(\mathbf{x})$, for any $\mathbf{x}\in\mathcal{X}$, and we notice that $\int\int L(y, f(\mathbf{x}))\rho_{\mathbf{x}}^T(\mathbf{x})dP_{y|\mathbf{x}}d\mathbf{x} = \int\int \phi(\mathbf{x})L(y, f(\mathbf{x}))\rho_{\mathbf{x}}^S(\mathbf{x})dP_{y|\mathbf{x}}d\mathbf{x}$. Inspired by this, the weighted version of (2) can be used, which leads to an importance ratio weighted (IRW) estimator that

$$\widetilde{f}^\phi := \operatorname*{argmin}_{f\in\mathcal{H}_K} \frac{1}{n}\sum_{i=1}^n \phi(\mathbf{x}_i^S)L\big(y_i^S, f(\mathbf{x}_i^S)\big) + \lambda\|f\|_K^2. \tag{3}$$

Throughout this paper, we focus on two types of the importance ratio that

(i) $\phi(\mathbf{x})$ is $\alpha$-uniformly bounded that is $\sup_{\mathbf{x}\in\mathcal{X}}\phi(\mathbf{x})\leq\alpha$, for some positive constant $\alpha$;

(ii) $\phi(\mathbf{x})$'s second moment is bounded that is $E_{\mathbf{x}\sim S}[\phi^2(\mathbf{x})]\leq\beta^2$, for some constant $\beta^2\geq 1$.

Note that case (i) reduces to the classical case without covariate shift if $\alpha = 1$. Yet, the bounded case is somewhat restrictive, and a much weaker condition is considered in case (ii). It is also clear that case (i) can be regarded as a special case of case (ii) by taking $\beta^2 = \alpha$. To see this, we have $E_{\mathbf{x}\sim S}[\phi(\mathbf{x})^2] = E_{\mathbf{x}\sim T}[\phi(\mathbf{x})]\leq\alpha$. It is interesting to notice that these two cases are related to Rényi divergence (Rényi, 1961) between $\rho_{\mathbf{x}}^S$ and $\rho_{\mathbf{x}}^T$. Specifically, the conditions in cases (i) and (ii) are equivalent to requiring $D_\infty(\rho_{\mathbf{x}}^T\|\rho_{\mathbf{x}}^S)$ and $D_2(\rho_{\mathbf{x}}^T\|\rho_{\mathbf{x}}^S)$ being bounded (Cortes et al., 2010), respectively. Moreover, as pointed out by Ma et al. (2023), (3) may result in significant inflation of the variance, largely due to the unbounded importance ratio. Thus, it is natural to consider a truncated importance ratio with a pre-specified threshold value $\gamma_n > 0$ (Ma et al., 2023; Gogolashvili et al., 2023). Precisely, we consider the following truncation

$$\phi_n(\mathbf{x}) := \min\{\phi(\mathbf{x}), \gamma_n\}.$$

Accordingly, the following truncated important ratio weighted (TIRW) estimator is considered that

$$\widehat{f}^\phi := \operatorname*{argmin}_{f\in\mathcal{H}_K} \frac{1}{n}\sum_{i=1}^n \phi_n(\mathbf{x}_i^S)L\big(y_i^S, f(\mathbf{x}_i^S)\big) + \lambda\|f\|_K^2, \tag{4}$$

where the theoretical suggestion of the choice of $\gamma_n$ is provided in Section 3.

## 3 Main results

In this section, we provide a unified theoretical analysis of the unweighted estimator in (2) and the TIRW estimator in (4) under two types of importance ratio. Let $\mathcal{H}_K$ denote the RKHS induced by a

symmetric, continuous, and positive semi-definite kernel function $K(\cdot, \cdot) : \mathcal{X} \times \mathcal{X} \rightarrow \mathcal{R}$. Under some regularity conditions, Mercer's theorem (Mercer, 1909) guarantees that $K$ has an eigen-expansion of the form

$$K(\mathbf{x}, \mathbf{x}') = \sum_{j=1}^{\infty} \mu_j \psi_j(\mathbf{x})\psi_j(\mathbf{x}'),$$

where $\{\mu_j\}_{j\geq 1}$ are the eigenvalues and $\{\psi_j\}_{j\geq 1}$ are the orthonormal eigenfunctions of $\mathcal{L}^2(\mathcal{X}, P_{\mathbf{x}}^T) = \{f : \int_{\mathcal{X}} f^2(\mathbf{x})\rho_{\mathbf{x}}^T(\mathbf{x})d\mathbf{x} < \infty\}$. We expand the kernel in $\mathcal{L}^2(\mathcal{X}, P_{\mathbf{x}}^T)$ for deriving bounds under the target distribution. For any $f \in \mathcal{H}_K$, we have $f = \sum_{j=1}^{\infty} a_j \psi_j$ with $a_j = \int_{\mathcal{X}} f(\mathbf{x})\psi_j(\mathbf{x})\rho_{\mathbf{x}}^T(\mathbf{x})d\mathbf{x}$ and the RKHS-norm is $\|f\|_K^2 = \sum_{j=1}^{\infty} a_j^2/\mu_j < \infty$. The kernel complexity function of $\mathcal{H}_K$ is then given as $R(\delta) = \sqrt{\frac{1}{n}\sum_{j=1}^{\infty}\min(\delta^2, \mu_j\|f^*\|_K^2)}$. In literature, $R(\delta)$ is important to quantify the localized Rademacher complexity of $\mathcal{H}_K$, which helps to build tight bounds in kernel-based methods (Mendelson, 2002; Koltchinskii & Yuan, 2010). For theoretical simplicity, we assume $\|f^*\|_K = 1$ in the rest of this paper, which is commonly considered in machine learning literature (Yang et al., 2017; Li et al., 2021). The following technical assumptions are required for the theoretical analysis.

**Assumption 1**: There exists some constant $\kappa > 0$ such that $\sup_{\mathbf{x}\in\mathcal{X}} |K(\mathbf{x}, \mathbf{x})| \leq \kappa$, and the eigenfunctions $\{\psi_j\}_{j\geq 1}$ are uniformly bounded such that $\sup_{\mathbf{x}\in\mathcal{X}} |\psi_j(\mathbf{x})| \leq 1$ for all $j \geq 1$.

**Assumption 2**: For some sufficiently small constant $u > 0$, we assume $E_I[L(y, f(\mathbf{x}))] - E_I[L(y, f^*(\mathbf{x}))] \geqslant c_0 \|f - f^*\|_I^2$ holds with some constant $c_0 > 0$, for all $\|f - f^*\|_I \leq u$, where $I \in \{S, T\}$.

Assumption 1 imposes a boundedness condition on the kernel function and its corresponding eigenfunctions, which is commonly considered in the literature and satisfied by many popularly used kernel functions with the compact support condition (Smale & Zhou, 2007; Steinwart et al., 2009; Mendelson & Neeman, 2010). Assumption 2 is a local $c_0$-strong convexity condition of the expected loss function with respect to $\mathcal{L}^2(\mathcal{X}, P_{\mathbf{x}}^S)$ and $\mathcal{L}^2(\mathcal{X}, P_{\mathbf{x}}^T)$ at $f^*$. Similar assumptions are also considered by Steinwart & Christmann (2008); Wainwright (2019); Li et al. (2021); Lian (2022). Note that many popularly used loss functions, including squared loss, check loss, Huber loss, logistic loss, and hinge loss, satisfy this assumption. More detailed discussions on Assumption 2 are deferred to Section A.7 of the supplementary material.

### 3.1 Convergence rate for the uniformly bounded case

We begin to establish the convergence rate of the unweighted estimator $\widehat{f}$ under the uniformly bounded case with covariate shift.

**Theorem 1.** *Under Assumptions 1-2, if the importance ratio is $\alpha$-uniformly bounded, let $\lambda > c_0\delta_n^2/4$ with $\delta_n$ being the smallest positive solution to $C\sqrt{\log n}R(\sqrt{\alpha}\delta) \leq c_0\delta^2/2$, then for some constant $c_1 > 0$, with probability at least $1 - n^{-c_1}$, we have*

$$\|\widehat{f} - f^*\|_T^2 \leq \alpha\left(\delta_n^2 + 2c_0^{-1}\lambda\right). \tag{5}$$

*Furthermore, based on* (5)*, we have*

$$\mathcal{E}_T^L(\widehat{f}) - \mathcal{E}_T^L(f^*) \leq c_L\alpha\left(\delta_n^2 + 2c_0^{-1}\lambda\right)^{1/2}. \tag{6}$$

Note that the existence and uniqueness of $\delta_n$ in Theorem 1 is guaranteed for any kernel class (Bartlett et al., 2005). To facilitate the comprehension of Theorem 1, we define an index as $d(\delta) = \min\{j \geq 1 | \mu_j \leq \delta^2\}$ for a given target error level $\delta > 0$. It is known that the kernel with eigenvalues satisfying that $\sum_{j=d(\delta)+1}^{\infty} \mu_j \leq Cd(\delta)\delta^2$ is referred to as the regular kernel class (Yang et al., 2017). From the definition of $d(\delta)$ and regular kernel, we can get that $\sum_{j=1}^{\infty}\min(\delta^2, \mu_j) \asymp d(\delta)\delta^2$, thus the inequality $C\sqrt{\log n}R(\sqrt{\alpha}\delta) \leq c_0\delta^2/2$ in Theorem 1 can be directly simplified as $\sqrt{\alpha \log n d(\sqrt{\alpha}\delta)/n} \leq C\delta$ under the assumption that $\|f^*\|_K = 1$. Consequently, we can rewrite (5) as

$$\|\widehat{f} - f^*\|_T^2 \leq C\alpha\left(\delta^2 + \alpha\frac{\log n}{n}d(\sqrt{\alpha}\delta)\right), \tag{7}$$

where $\delta$ satisfies $\alpha \frac{\log n}{n} d(\sqrt{\alpha}\delta) \leq C\delta^2$ and $\lambda \asymp \delta^2$ (see Section C.4 of the supplementary material for the detailed proof). The bound in (7) controls a type of bias-variance tradeoff by the choice of $\delta$, and hence $\lambda$. Under the uniformly bounded case, Ma et al. (2023) has established a minimax rate of the order $\alpha \inf_{\delta>0}\{\delta^2 + \frac{\sigma^2 d(\sqrt{\alpha}\delta)}{n}\}$ for the squared loss function. It is worthnoting that the result in (7) can also attain this lower bound up to a logarithmic factor under some weaker conditions. Particularly, the assumption of Ma et al. (2023) that the noise terms are sub-Gaussian is no longer needed in (7). Although the convergence rate of (6) is sub-optimal for general loss function $L$ considered in Section 2.1, it becomes optimal at the rate of $c_0'\alpha\left(\delta_n^2 + 2c_0^{-1}\lambda\right)$ if the following assumption holds.

**Assumption 3**: For some sufficiently small constant $u > 0$, we assume $E_T\left[L(y, f(\mathbf{x}))\right] - E_T\left[L\left(y, f^*(\mathbf{x})\right)\right] \leq c_0'\|f - f^*\|_T^2$ holds with some constant $c_0' > c_0 > 0$, for all $\|f - f^*\|_T \leq u$.

In fact, Assumption 3 is a mild condition that can be satisfied by many commonly used losses. For instance, for the squared loss, the equality always holds with $c_0' = 1$; for the check loss, Assumption 3 is satisfied if the conditional density of the noise term is uniformly bounded (Zhang et al., 2021; Lian, 2022). Moreover, the regular kernel class includes kernels with either finite rank (i.e., linear or polynomial kernels), polynomially decayed eigenvalues (i.e., Sobolev kernels), or exponentially decayed eigenvalues (i.e., Gaussian kernels). Corollary 1 provides the convergence rate of $\widehat{f}$ over these three specific kernel classes.

**Corollary 1.** *Under Assumptions 1-3, if the kernel has a finite rank $D$, and let $\lambda = C\frac{\alpha D \log n}{n}$, then with probability at least $1 - n^{-c_1}$, we have*

$$\|\widehat{f} - f^*\|_T^2 \asymp \mathcal{E}_T^L(\widehat{f}) - \mathcal{E}_T^L(f^*) \leq C\frac{\alpha^2 D \log n}{n}, \tag{8}$$

*and if the eigenvalues of the kernel decay polynomially such as $\mu_j \leq Cj^{-2r}$ with some constant $r > 1/2$ for $j = 1, 2, \ldots$, and let $\lambda = C\alpha^{\frac{2r-1}{2r+1}}(\frac{\log n}{n})^{\frac{2r}{2r+1}}$, then with probability at least $1 - n^{-c_1}$, we have*

$$\|\widehat{f} - f^*\|_T^2 \asymp \mathcal{E}_T^L(\widehat{f}) - \mathcal{E}_T^L(f^*) \leq C\left(\frac{\alpha^2 \log n}{n}\right)^{\frac{2r}{2r+1}}, \tag{9}$$

*and if the eigenvalues of the kernel decay exponentially such as $\mu_j \asymp e^{-Cj\log j}$, and let $\lambda = C\frac{\alpha \log^2 n}{n}$, then with probability at least $1 - n^{-c_1}$, we have*

$$\|\widehat{f} - f^*\|_T^2 \asymp \mathcal{E}_T^L(\widehat{f}) - \mathcal{E}_T^L(f^*) \leq C\frac{\alpha^2 \log^2 n}{n}, \tag{10}$$

Note that these bounds in (8)-(10) reduce to the known minimax lower bounds (Yang et al., 2017) for the squared loss without covariate shift (i.e., $\alpha = 1$). As the uniform boundedness implies the boundedness of the second moment, the convergence rate of the TIRW estimator $\widehat{f}^\phi$ for the uniformly bounded case is similar to Theorem 3 in the next section by replacing $\alpha$ with $\beta^2$.

### 3.2 Convergence rate for the second moment bounded case

The optimality for the $\alpha$-uniformly bounded importance ratio condition relies on the inequality that $\|\widehat{f} - f^*\|_T^2 \leq \alpha\|\widehat{f} - f^*\|_S^2$. Yet, such a desired relation is not guaranteed for the second moment bounded case. Theorem 2 shows that the unweighted estimator $\widehat{f}$ is still consistent, but not optimal.

**Theorem 2.** *Under Assumptions 1-2, if the importance ratio satisfies that $E_{\mathbf{x}\sim S}[\phi^2(\mathbf{x})] \leq \beta^2$, let $\lambda > c_0\delta_n^2/4$ with $\delta_n$ being the smallest positive solution to $C\sqrt{\log n}R((c_0^{-1}c_L\sqrt{\beta^2}\delta)^{1/2}) \leq c_0\delta^2/2$, then for some constant $c_2 > 0$, with probability at least $1 - n^{-c_2}$, we have*

$$\|\widehat{f} - f^*\|_T^2 \leq c_0^{-1}c_L\sqrt{\beta^2}\left(\delta_n^2 + 2c_0^{-1}\lambda\right)^{1/2}. \tag{11}$$

*Furthermore, based on* (11)*, we have*

$$\mathcal{E}_T^L(\widehat{f}) - \mathcal{E}_T^L(f^*) \leq c_L\sqrt{\beta^2}\left(\delta_n^2 + 2c_0^{-1}\lambda\right)^{1/2}. \tag{12}$$

To see the bound in (11) is sub-optimal, we consider the kernel with finite rank $D$, we can show that $\delta_n \asymp (\frac{\sqrt{\beta^2 D \log n}}{n})^{1/3}$, and if we take $\lambda \asymp (\frac{\sqrt{\beta^2 D \log n}}{n})^{2/3}$, the unweighted estimator satisfies that $\|\widehat{f} - f^*\|_T^2 = O_P((\frac{\beta^4 D \log n}{n})^{1/3})$, which is far from the optimal rate (see Section C.4 of the supplementary material for examples of finite rank, polynomially and exponentially decay kernel classes). To deal with the sub-optimality, we consider the importance ratio correction ensuring that $E_S[\phi(\mathbf{x})L(y, f(\mathbf{x}))] = E_T[L(y, f(\mathbf{x}))]$. The following theorem shows that the TIRW estimator $\widehat{f}^\phi$ can again reach a sharp convergence rate up to logarithmic factors under some mild conditions.

**Theorem 3.** *Under Assumptions 1-2, if the importance ratio satisfies that $E_{\mathbf{x} \sim S}[\phi^2(\mathbf{x})] \leq \beta^2$, let $\lambda > c_0 \delta_n^2/4$ with $\delta_n$ being the smallest positive solution to $C\sqrt{\beta^2} \log n R(\delta) \leq c_0 \delta^2/2$, and set the truncation level $\gamma_n = \sqrt{n\beta^2}$, then for some constant $c_3 > 0$, with probability at least $1 - n^{-c_3}$, we have*

$$\|\widehat{f}^\phi - f^*\|_T^2 \leq \delta_n^2 + 2c_0^{-1}\lambda. \tag{13}$$

*Furthermore, based on* (13)*, we have*

$$\mathcal{E}_T^L(\widehat{f}^\phi) - \mathcal{E}_T^L(f^*) \leq \frac{1}{2}c_0 \delta_n^2 + 2\lambda. \tag{14}$$

Similar to (7), when dealing with regular kernel classes, the bounds in (13) and (14) become

$$\|\widehat{f}^\phi - f^*\|_T^2 \asymp \mathcal{E}_T^L(\widehat{f}^\phi) - \mathcal{E}_T^L(f^*) \leq C\Big(\delta^2 + \beta^2 \frac{\log^2 n}{n} d(\delta)\Big), \tag{15}$$

where $\delta$ is any solution of $\beta^2 \frac{\log^2 n}{n} d(\delta) \leq C\delta^2$ and $\lambda \asymp \delta^2$. Since the second moment boundedness can be implied by the uniform boundedness, it can be concluded that $\widehat{f}^\phi$ also reaches the minimax lower bound for the square loss up to logarithmic factors (with $\alpha$ substituted by $\beta^2$). For three specific kernel classes, we also have the following Corollary for the TIRW estimator $\widehat{f}^\phi$.

**Corollary 2.** *Under Assumptions 1-2, if the kernel has a finite rank $D$, and let $\lambda = C\frac{\beta^2 D \log^2 n}{n}$, then with probability at least $1 - n^{-c_3}$, we have*

$$\|\widehat{f}^\phi - f^*\|_T^2 \asymp \mathcal{E}_T^L(\widehat{f}^\phi) - \mathcal{E}_T^L(f^*) \leq C\frac{\beta^2 D \log^2 n}{n}, \tag{16}$$

*and if the eigenvalues of the kernel decay polynomially such as $\mu_j \leq Cj^{-2r}$ with a constant $r > 1/2$ for $j = 1, 2, \ldots$, and let $\lambda = C(\frac{\beta^2 \log n}{n})^{\frac{2r}{2r+1}}$, then with probability at least $1 - n^{-c_3}$, we have*

$$\|\widehat{f}^\phi - f^*\|_T^2 \asymp \mathcal{E}_T^L(\widehat{f}^\phi) - \mathcal{E}_T^L(f^*) \leq C\left(\frac{\beta^2 \log^2 n}{n}\right)^{\frac{2r}{2r+1}}, \tag{17}$$

*and if the eigenvalues of the kernel decay exponentially such as $\mu_j \asymp e^{-Cj \log j}$, and let $\lambda = C\frac{\beta^2 \log^3 n}{n}$, then with probability at least $1 - n^{-c_3}$, we have*

$$\|\widehat{f} - f^*\|_T^2 \asymp \mathcal{E}_T^L(\widehat{f}) - \mathcal{E}_T^L(f^*) \leq C\frac{\beta^2 \log^3 n}{n}, \tag{18}$$

For easy reference, we summarize some important theoretical results that have been established by us in table 1.

Table 1: Established convergence rates under different cases

| Kernel class | Uniformly bounded case | | Moment bounded case | |
| --- | --- | --- | --- | --- |
| | Unweighted estimator | TIRW estimator | Unweighted estimator | TIRW estimator |
| Finite rank $D$ | $O_P(\frac{\alpha^2 D \log n}{n})$ | $O_P(\frac{\alpha D \log^2 n}{n})$ | $O_P((\frac{\beta^4 D \log n}{n})^{1/3})$ | $O_P(\frac{\beta^2 D \log^2 n}{n})$ |
| Polynomial decay | $O_P((\frac{\alpha^2 \log n}{n})^{\frac{2r}{2r+1}})$ | $O_P((\frac{\alpha \log^2 n}{n})^{\frac{2r}{2r+1}})$ | $O_P((\frac{\beta^4 \log n}{n})^{\frac{2r}{6r+1}})$ | $O_P((\frac{\beta^2 \log^2 n}{n})^{\frac{2r}{2r+1}})$ |
| Exponential decay | $O_P(\frac{\alpha^2 \log^2 n}{n})$ | $O_P(\frac{\alpha \log^3 n}{n})$ | $O_P((\frac{\beta^4 \log^2 n}{n})^{1/3})$ | $O_P(\frac{\beta^2 \log^3 n}{n})$ |

# 4    Numerical experiments

In this section, we validate our theoretical analyses by performing numerical experiments on both synthetic data and real applications. To estimate the importance ratios, we apply the Kullback-Leibler importance estimation procedure (KLIEP) (Sugiyama et al., 2007b), and the estimation details are provided in the supplementary material. For brevity, we only report the performance of kernel-based quantile regression (KQR) where the check loss is used in synthetic data analysis, and the performance of kernel support vector support machine (KSVM) with hinge loss in the real data analysis; completed results for other loss functions as well as the detailed settings and results for the other examples, including for the multi-dimensional cases, can be found in Section A of the supplementary material. In our experiments, we consider the RKHS induced by Gaussian kernel in all the examples, and all the experiments are replicated 100 times in the synthetic data analysis.

## 4.1    Synthetic data analysis

We investigate the performance of KQR under covariate shift with the following generating model

$$ y = f_0(x) + \left(1 + r\left(x - 0.5\right)^2\right)\sigma(\varepsilon - \Phi^{-1}(\tau)), $$

where $f_0(x) = \sin(\pi x)$ with $x \in \mathcal{R}$, $\Phi$ denotes the CDF function of the standard normal distribution and $\varepsilon \sim N(0,1)$. We consider $\rho_{\mathbf{x}}^S \sim N(\mu_1, \sigma_1^2)$ and $\rho_{\mathbf{x}}^T \sim N(\mu_2, \sigma_2^2)$ with $\mu_1 = 0, \sigma_1^2 = 0.4, \mu_2 = 0.5, \sigma_2^2 = 0.3$ for the uniformly bounded case and $\mu_1 = 0, \sigma_1^2 = 0.3, \mu_2 = 1, \sigma_2^2 = 0.5$ for the moment bounded case, respectively. Moreover, we set $r = 0$ and $\sigma = 0.5$ for the homoscedastic case, and $r = 1$ and $\sigma = 0.3$ for the heteroscedastic case, respectively. Note that the simulation results for the case that $\tau = 0.3$ and $r = 1$ are presented in Figure 2 under the uniformly bounded case and in Figure 3 under the moment bounded case; completed results for other cases are provided in the supplementary material. We compare the averaged mean square error (MSE) and empirical excess risk of the unweighted estimator and weighted estimator, either with true or estimated weights across different choices of regularization parameter $\lambda$, source sample size $n$, and target sample size $m$.

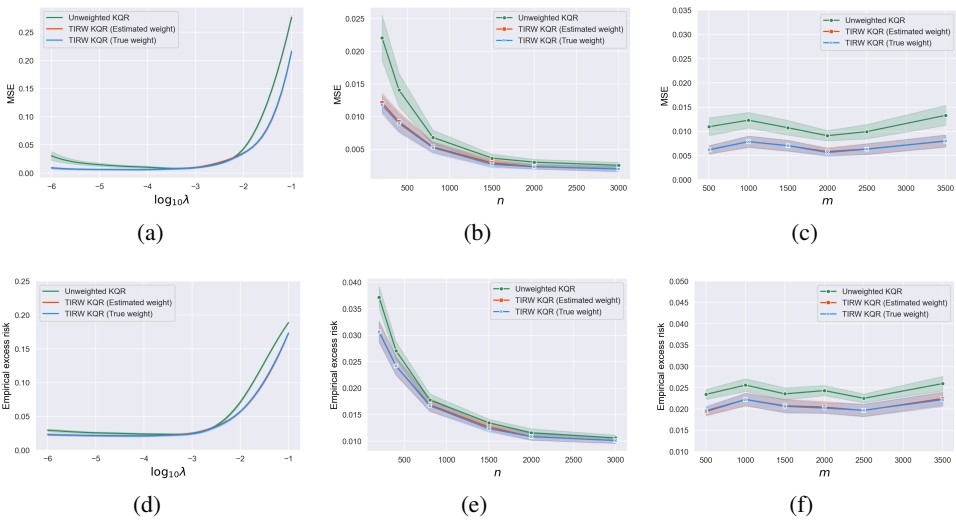

Figure 2: Averaged MSE and empirical excess risk for unweighted KQR, TIRW KQR with true weight and estimated weight, respectively. Note that in (a) and (d), the curves are plotted with respect to $\log_{10} \lambda$ with $n = 500, m = 1000$; in (b) and (e) the curves are plotted with respect to $n$ with fixed $m = 1000, \lambda = 10^{-4}$; in (c) and (f), the curves are plotted with respect to $m$ with fixed $n = 500, \lambda = 10^{-4}$.

From (a) and (d) in Figure 2, we can conclude that the error of the unweighted estimator is very close to that of the weighted estimator for the uniformly bounded case, which is consistent with our theoretical findings in Section 3. For the moment bounded case as demonstrated in (a) and (d) of Figure 3, the weighted estimator consistently outperforms its unweighted counterpart for all choices of $\lambda$. Even when $\lambda$ is far from its optimal choice, the weighted estimator still maintains a lower

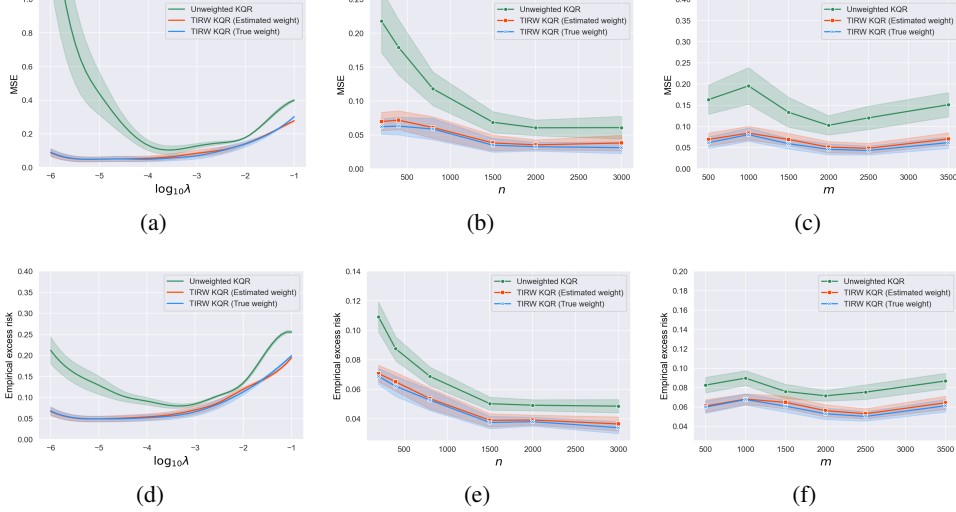

Figure 3: Averaged MSE and empirical excess risk for unweighted KQR, TIRW KQR with true weight and estimated weight, respectively. Note that in (a) and (d), the curves are plotted with respect to $\log_{10} \lambda$ with $n = 500, m = 1000$; in (b) and (e) the curves are plotted with respect to $n$ with fixed $m = 1000, \lambda = 10^{-4}$; in (c) and (f), the curves are plotted with respect to $m$ with fixed $n = 500, \lambda = 10^{-4}$.

error level with significant improvement over the unweighted estimator whose error is on occasion extremely high for small choices of $\lambda$. Additionally, it is clear from Panels (b) and (e) of Figures 2 and 3 that in the moment bounded case, the error curves for the unweighted estimator have some significant gaps with those of the weighted estimator, while in the bounded case, these curves tend to coincide as $n$ grows. It is also clear from the table 1 that the trends in Figure 2 (b)-(e) and Figure 3 (b)-(e) almost agree with the explicit convergence rate where the Gaussian kernel with exponential decay is used. This phenomenon is consistent with our theoretical conclusion that the unweighted estimator can only achieve sub-optimal rates when the importance ratio is moment bounded and attain optimal rates when uniformly bounded. Finally, we note that the target sample size has a subtle influence on all the estimators, as demonstrated in Panels (c) and (f) of Figures 2 and 3.

## 4.2 Real case study

We consider the binary classification problem on the Raisin dataset, which is available in `https://archive.ics.uci.edu/ml/datasets.php`. The dataset contains 900 instances and 7 attributes. After standardization, the data are first randomly split into source and target datasets. We introduce a binary random variable $s$ that serves as a labeling indicator and assign it to the source dataset if $s = 0$. To ensure that the conditional distribution of different datasets remains invariant, we require $s$ to be conditionally independent with the response $y$ given the covariate $\mathbf{x}$ (Zadrozny, 2004; Quinonero-Candela et al., 2008). We implement KSVM in which the covariate shift only exists in the first covariate. Specifically, for $\ell > 0$, we conduct the splitting rule as $P(s_i = 1 \mid \mathbf{x}_i) = \min(1, (x_{i,0} - c)^2/\ell)$, where $x_{i,0}$ is the first element of $\mathbf{x}_i$ and $c = \min_i\{x_{i,0}\}$. We refer to $\ell$ as the shift level that a smaller value of $\ell$ favors a greater distinction between source and target datasets. We set the turning parameter $C_\lambda = (n\lambda)^{-1}$. For the TIRW estimator, we use importance weighted cross validation (IWCV) (Sugiyama et al., 2007a) to tune the truncation parameter $\gamma_n$. We refer to Section A in the supplementary material for the details of the adopted method.

Figure 4 summarizes the accuracy rate and standard error in different settings. As shown in this figure, the unweighted estimator has poor performance on target data, even for the optimal choice of $C_\lambda$. Nevertheless, the weighted estimator has a significant improvement in performance. Surprisingly, the weighted estimator is relatively stable with respect to the choices of $C_\lambda$, which is consistent with our results on synthetic data. Moreover, we can find that a proper truncation slightly improves the accuracy rate of the weighted estimator while reducing its variation.

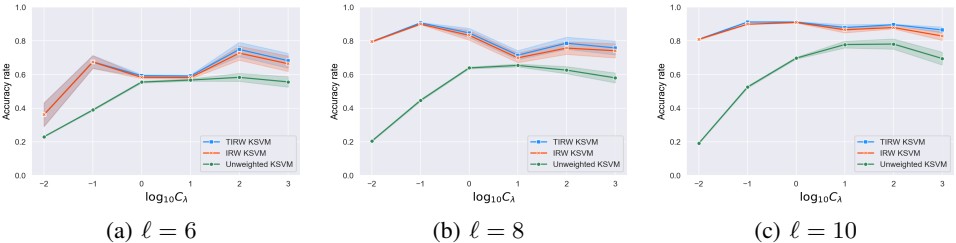

(a) $\ell = 6$        (b) $\ell = 8$        (c) $\ell = 10$

Figure 4: Averaged accuracy rate for unweighted KSVM, IRW KSVM and TIRW KSVM with different shift levels $\ell$, respectively.

## 5 Conclusion

In this work, we propose a unified analyzing framework for kernel-based methods under covariate shift and show that the predictive function can be better estimated by using the distribution information of the covariates from both aspects of theory and numerical performance. The established theoretical results fulfill the gap in understanding the influence of the covariate shift under various kernel-based learning problems, both researchers and practitioners may benefit from our theoretical findings. Extensive numerical studies also provide empirical evidence and confirm the established theoretical results. Note that it is very interesting to test if there exists a distribution shift in practical applications and if the distribution shift satisfies the uniformly bounded or moment bounded assumptions. Unfortunately, the relevant approaches have remained lacking to our best knowledge. And it is interesting to note that as shown in our real applications, the TIRW estimator always outperforms the unweighted estimator, and thus we suggest using the TIRW estimator to analyze the real-life dataset. Moreover, in the theory, we directly use the true importance ratio, and it is difficult to derive the consistency of the estimated ratio and plug it into our theoretical analysis. In the machine learning literature, there exist several measures to quantify the divergence between two distributions, including but not limited to $f$-divergence, Wasserstein distance and kernel mean matching. It is still an open problem if it still works when we use other measures rather than the importance ratio. We leave such promising and interesting topics as potential future work.

## Acknowledgment

Xingdong Feng's research is supported in part by NSFC-12371270 and Shanghai Science and Technology Development Funds (23JC1402100). Xin He's research is supported in part by NSFC-11901375 and Program for Innovative Research Team of Shanghai University of Finance and Economics. This research is also supported by Shanghai Research Center for Data Science and Decision Technology and the Fundamental Research Funds for the Central Universities.

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
