# Supplementary material for "Towards a Unified Analysis of Kernel-based Methods Under Covariate Shift"

The supplemental material is organized as follows. Section A provides the results of all the additional synthetic experiments and real data results for various kernel-based methods and the detailed settings. Section B describes the algorithm details we use in Section A. In Section C, we provide some useful lemmas and all the technical proofs of the theoretical results in the main text. The python code implementing the proposed method is available at `https://github.com/WangCaixing-96/Kernel_CS`.

## A Additional numerical results

In this section, we provide more experiment results, including KRR (Section A.1), KQR for various $\tau$ and $r$ (Section A.2), kernel logistic regression (KLR) (Section A.3) and multi-source real data studies (Section A.4), that further confirm our theoretical findings. Section A.5 and Section A.6 are devoted to introducing KLIEP and IWCV. We also add some discussion on Assumption 2 of the main text in Section A.7.

### A.1 Kernel ridge regression

For the squared loss, we consider the following two examples.

**Example S1**: The response $y$ is generated by $y = f_0(x) + \sigma\varepsilon$, where $f_0(x) = e^{-\frac{1}{x^2}}$ and $\varepsilon \sim N(0, 1)$. The source and target distributions are $\rho_{\mathbf{x}}^S(\mathbf{x}) \sim N(\mu_1, \sigma_1^2)$ and $\rho_{\mathbf{x}}^T(\mathbf{x}) \sim N(\mu_2, \sigma_2^2)$ with $\mu_1 = 0, \mu_2 = 0.8, \sigma_1^2 = 0.5, \sigma_2^2 = 0.3$ for the uniformly bounded case and $\mu_1 = 0, \mu_2 = 1.5, \sigma_1^2 = 0.3, \sigma_2^2 = 0.5$ for the moment bounded case, respectively. The noise level $\sigma$ is set to 0.05 for both uniformly bounded and moment bounded cases. The results are shown in Figure 1

**Example S2**: The response $y$ is generated by $y = f_0(\mathbf{x}) + \sigma\varepsilon$, where $\mathbf{x} = (x_0, x_1, x_2)^\top \in \mathcal{R}^3, f_0(\mathbf{x}) = \sin(2\pi x_0) - e^{-x_1^2 - x_2^2}$ and $\varepsilon \sim N(0, 1)$. Let $g(x_0; \alpha, \beta)$ denote the probability density function of Beta distribution with parameters $\alpha, \beta$. It is easy to see that the importance ratio $\phi(\mathbf{x})$ for this case is not uniformly bounded but second moment bounded if and only if $\alpha_t < \alpha_s, 2\alpha_t \geq \alpha_s, 2\beta_t \geq \beta_s$ or $\beta_t < \beta_s, 2\alpha_t \geq \alpha_s, 2\beta_t \geq \beta_s$. In the 3-dimensional KRR experiment, we consider $\rho_{\mathbf{x}}^S(\mathbf{x}) = g(x_0; \alpha_s, \beta_s)$ and $\rho_{\mathbf{x}}^T(\mathbf{x}) = g(x_0; \alpha_t, \beta_t)$ with $\alpha_s = 2.5, \beta_s = 1.5, \alpha_t = 3, \beta_t = 4$ for the uniformly bounded case and $\alpha_s = 4, \beta_s = 1, \alpha_t = 3, \beta_t = 6$ for the moment bounded case, respectively. The noise level $\sigma$ is set to 0.3 for both uniformly bounded and moment bounded cases. The results are shown in Figure 2

From (d) in Figures 1 and 2, we observe that for the moment bounded case, the TIRW estimator has a great improvement compared to the unweighted estimator, even for the choice of $\lambda$ that is far away from the optimum. Nevertheless, for the bounded case, we can see from (a) in Figure 1 and Figure 2 that there has a negligible gap between the performance of the unweighted estimator and that of the TIRW estimator as long as we choose $\lambda$ that is close to optimum. For the poor choice of $\lambda$, the TIRW estimator still performs significantly better. From (b) and (e) in Figures 1 and 2, it is shown that the error curve has an explicit gap with those of weighted estimators for the moment bounded case, whereas it is very close for the uniformly bounded case. From (c) and (f) in Figures 1 and 2, we observe that the target data size $m$ has a subtle influence on our estimators.

37th Conference on Neural Information Processing Systems (NeurIPS 2023).

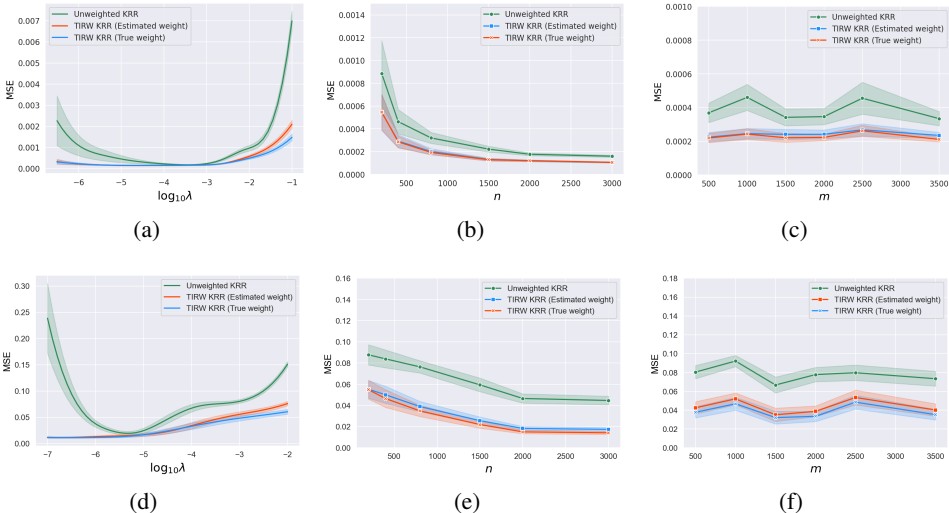

Figure 1: Average MSE for unweighted KRR, TIRW KRR with true weight and estimated weight, respectively (The top panel is for the bounded case and the bottom is for the moment bounded case; in (a) and (d), the curves are plotted with respect to $\log_{10} \lambda$ with $n = 500, m = 1000$; in (b) and (e) the curves are plotted with respect to $n$ with fixed $m = 1000, \lambda = 10^{-4}$; in (c) and (f), the curves are plotted with respect to $m$ with fixed $n = 500, \lambda = 10^{-4}$)

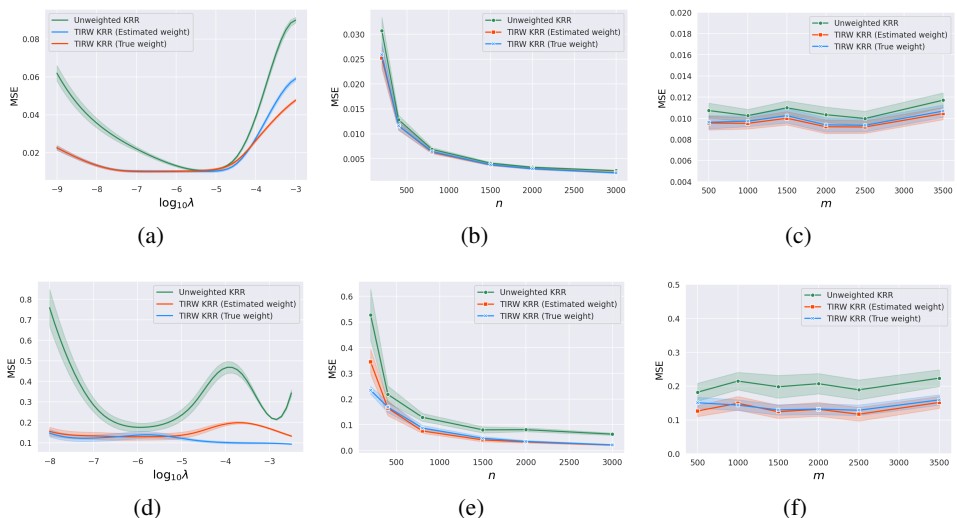

Figure 2: Average MSE for unweighted KRR, TIRW KRR with true weight and estimated weight, respectively (The top panel is for the bounded case and the bottom is for the moment bounded case; in (a) and (d), the curves are plotted with respect to $\log_{10} \lambda$ with $n = 500, m = 1000$; in (b) and (e) the curves are plotted with respect to $n$ with fixed $m = 1000, \lambda = 5 \times 10^{-5}$; in (c) and (f), the curves are plotted with respect to $m$ with fixed $n = 500, \lambda = 5 \times 10^{-5}$)

## A.2 Kernel quantile regression

For the check loss, we consider the following two examples.

**Example S3**: This example continues to study the KQR with a 1-dimensional covariate under the same setting as in the main text. Here we further conduct experiments for various combinations of $\tau \in \{0.3, 0.5, 0.7\}$ and $r \in \{0, 1\}$.

**Example S4**: The response $y$ is generated by $y = f_0(\mathbf{x}) + (1 + rx_0)\sigma(\varepsilon - t_4^{-1}(\tau))$, $\mathbf{x} = (x_0, x_1, x_2)^\top \in \mathcal{R}^3$, where $f_0(\mathbf{x}) = \sin(1.5\pi x_0) - e^{-x_1^2 - x_2^2}$, $t_4$ denotes the CDF function of t-distribution with $4$ degrees of freedom and $\varepsilon \sim t_4$. We consider $\rho_{\mathbf{x}}^S(\mathbf{x}) = g(x_0; \alpha_s, \beta_s)$ and $\rho_{\mathbf{x}}^T(\mathbf{x}) = g(x_0; \alpha_t, \beta_t)$ with $\alpha_s = 2.5, \beta_s = 1.5, \alpha_t = 3, \beta_t = 6$ for the uniformly bounded case and $\alpha_s = 5.5, \beta_s = 1.5, \alpha_t = 3, \beta_t = 6$ for the moment bounded case, respectively. We set $r = 0$ and $\sigma = 0.3$ for the homoscedastic case and $r = 1$ and $\sigma = 0.3$ for the heteroscedastic case, respectively.

Note that the numerical results are provided in Sections A.2.1–A.2.4. Specifically, Figures 3 in Section A.2.1 present the results for the uniformly bounded case in Example S3, and the results for the moment bounded case in Example S3 are presented in Figure 4 of Section A.2.2. Moreover, Figure 5 in Section A.2.3 presents the results for the uniformly bounded case in Example S4 with various combinations of $\tau \in \{0.3, 0.5, 0.7\}$ and $r \in \{0, 1\}$, and the results for the moment bounded case in Example S4 are presented in Figure 6 of Section A.2.4.

It is thus clear that the TIRW estimator is robust for different combinations of $\tau$ and $r$. Specifically, the TIRW estimator takes a significant advantage over the unweighted estimator when the important ratio is indeed unbounded. Nevertheless, for the bounded case, the TIRW estimator seems to be not necessary since if the choice of turning parameter is nearly-optimal or the source data size is relatively enough, there is a negligible gap between the TIRW estimator and the unweighted estimator.

## A.2.1 Uniformly bounded case in Example S3

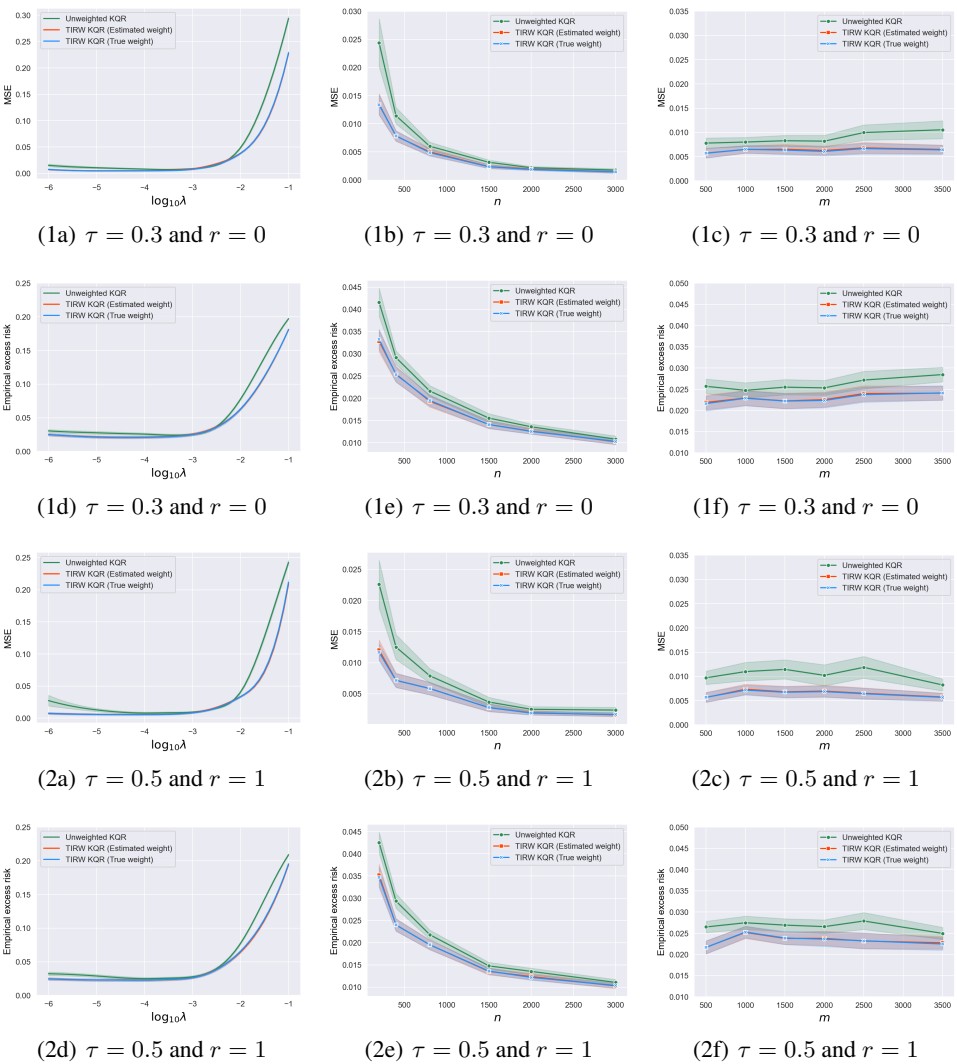

(1a) $\tau = 0.3$ and $r = 0$    (1b) $\tau = 0.3$ and $r = 0$    (1c) $\tau = 0.3$ and $r = 0$

(1d) $\tau = 0.3$ and $r = 0$    (1e) $\tau = 0.3$ and $r = 0$    (1f) $\tau = 0.3$ and $r = 0$

(2a) $\tau = 0.5$ and $r = 1$    (2b) $\tau = 0.5$ and $r = 1$    (2c) $\tau = 0.5$ and $r = 1$

(2d) $\tau = 0.5$ and $r = 1$    (2e) $\tau = 0.5$ and $r = 1$    (2f) $\tau = 0.5$ and $r = 1$

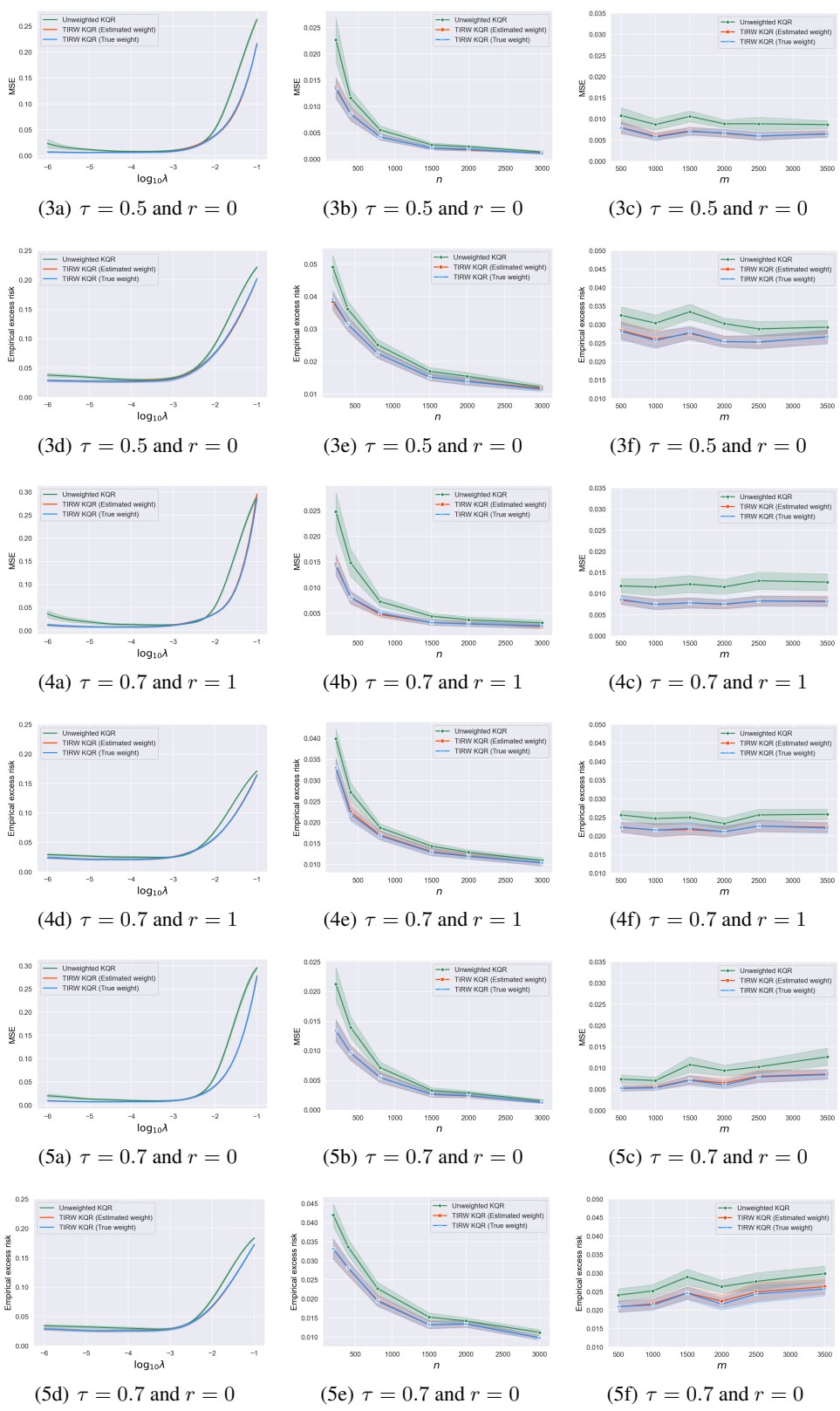

Figure 3: Average MSE and empirical excess risk for unweighted KQR, TIRW KQR with true weight and estimated weight, respectively (in the left panel, the curves are plotted with respect to $\log_{10}\lambda$ with $n = 500, m = 1000$; in the middle panel, the curves are plotted with respect to $n$ with fixed $m = 1000, \lambda = 10^{-4}$; in the right panel, the curves are plotted with respect to $m$ with fixed $n = 500, \lambda = 10^{-4}$)

## A.2.2 Moment bounded case in Example S3

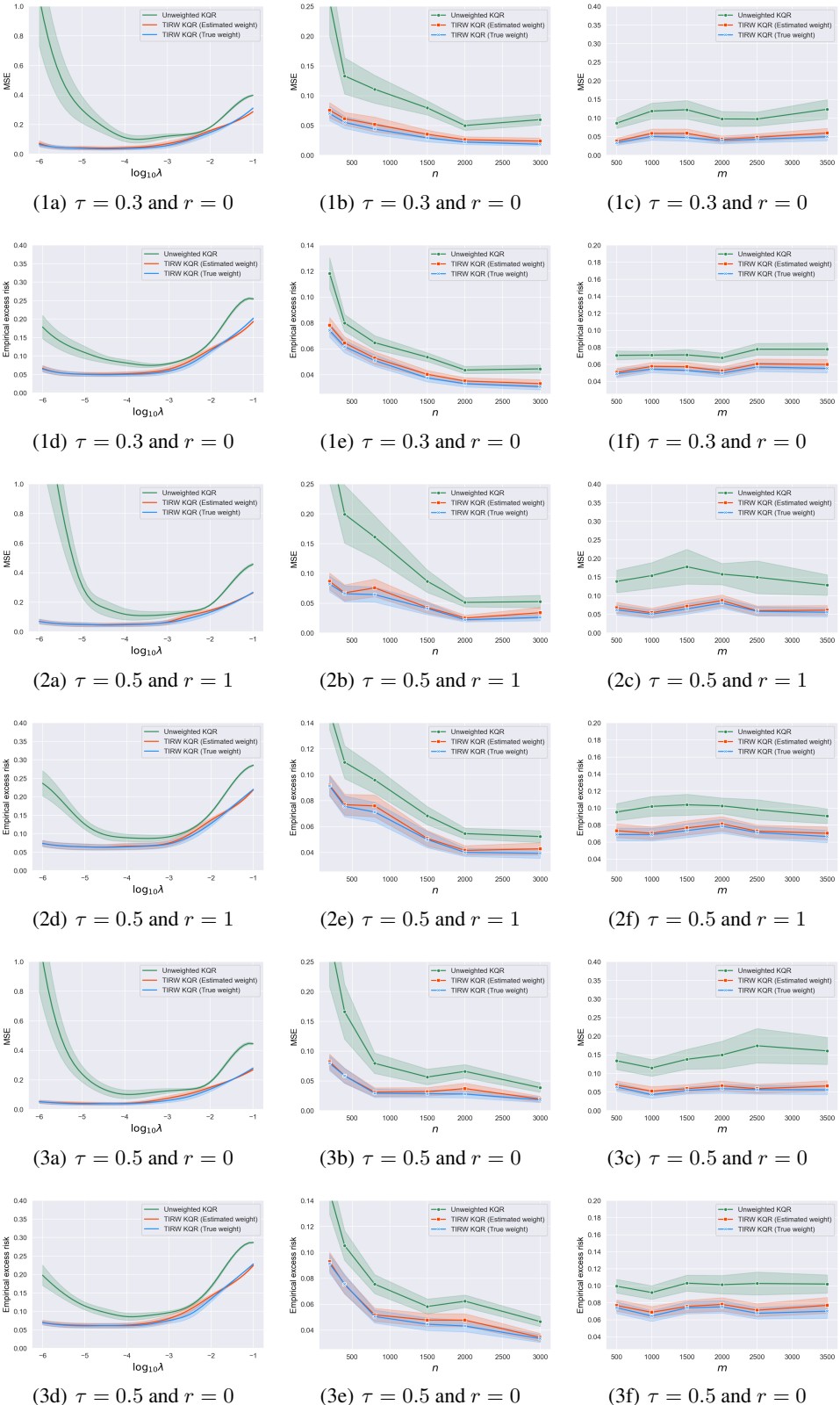

(1a) $\tau = 0.3$ and $r = 0$     (1b) $\tau = 0.3$ and $r = 0$     (1c) $\tau = 0.3$ and $r = 0$

(1d) $\tau = 0.3$ and $r = 0$     (1e) $\tau = 0.3$ and $r = 0$     (1f) $\tau = 0.3$ and $r = 0$

(2a) $\tau = 0.5$ and $r = 1$     (2b) $\tau = 0.5$ and $r = 1$     (2c) $\tau = 0.5$ and $r = 1$

(2d) $\tau = 0.5$ and $r = 1$     (2e) $\tau = 0.5$ and $r = 1$     (2f) $\tau = 0.5$ and $r = 1$

(3a) $\tau = 0.5$ and $r = 0$     (3b) $\tau = 0.5$ and $r = 0$     (3c) $\tau = 0.5$ and $r = 0$

(3d) $\tau = 0.5$ and $r = 0$     (3e) $\tau = 0.5$ and $r = 0$     (3f) $\tau = 0.5$ and $r = 0$

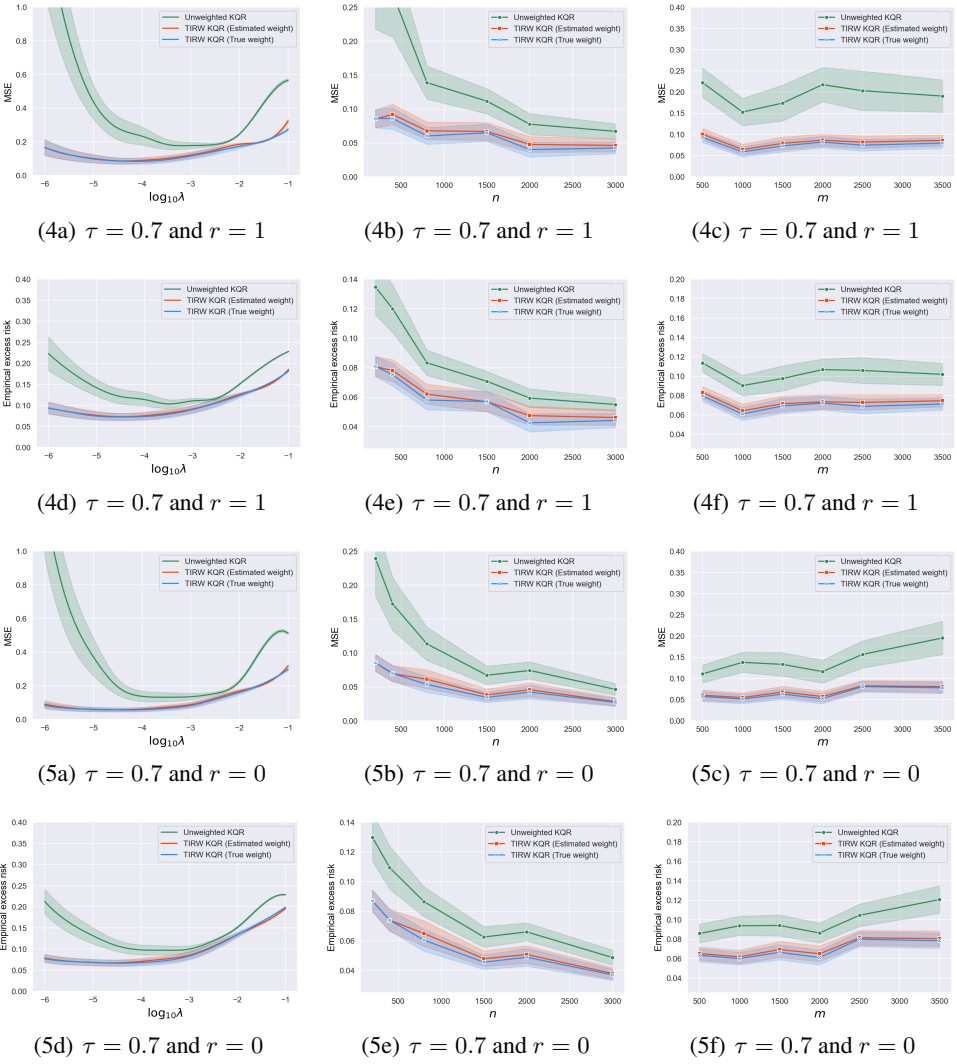

Figure 4: Average MSE and empirical excess risk for unweighted KQR, TIRW KQR with true weight and estimated weight, respectively (in the left panel, the curves are plotted with respect to $\log_{10} \lambda$ with $n = 500, m = 1000$; in the middle panel, the curves are plotted with respect to $n$ with fixed $m = 1000, \lambda = 10^{-4}$; in the right panel, the curves are plotted with respect to $m$ with fixed $n = 500, \lambda = 10^{-4}$)

## A.2.3 Uniformly bounded case in Example S4

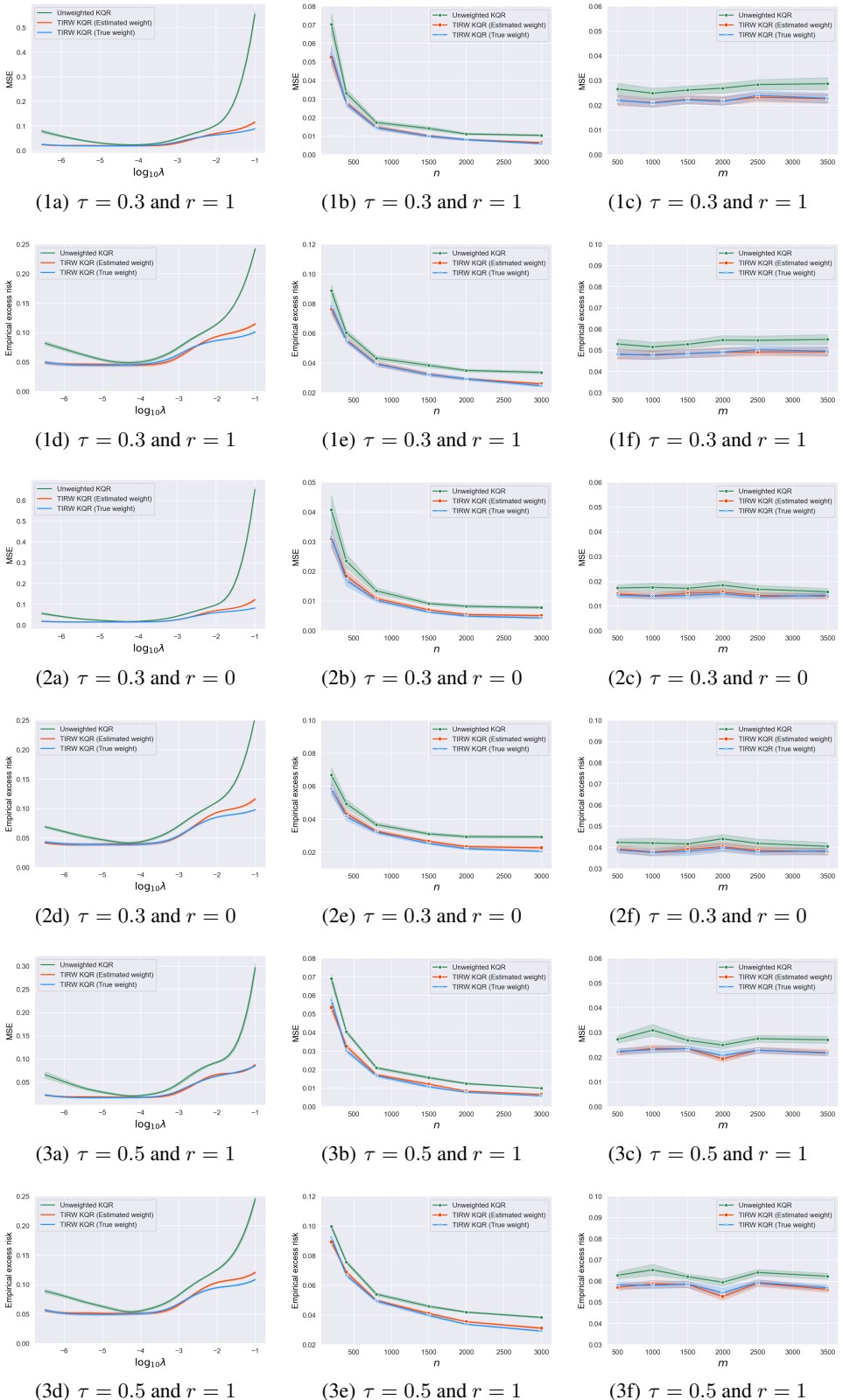

(1a) $\tau = 0.3$ and $r = 1$    (1b) $\tau = 0.3$ and $r = 1$    (1c) $\tau = 0.3$ and $r = 1$

(1d) $\tau = 0.3$ and $r = 1$    (1e) $\tau = 0.3$ and $r = 1$    (1f) $\tau = 0.3$ and $r = 1$

(2a) $\tau = 0.3$ and $r = 0$    (2b) $\tau = 0.3$ and $r = 0$    (2c) $\tau = 0.3$ and $r = 0$

(2d) $\tau = 0.3$ and $r = 0$    (2e) $\tau = 0.3$ and $r = 0$    (2f) $\tau = 0.3$ and $r = 0$

(3a) $\tau = 0.5$ and $r = 1$    (3b) $\tau = 0.5$ and $r = 1$    (3c) $\tau = 0.5$ and $r = 1$

(3d) $\tau = 0.5$ and $r = 1$    (3e) $\tau = 0.5$ and $r = 1$    (3f) $\tau = 0.5$ and $r = 1$

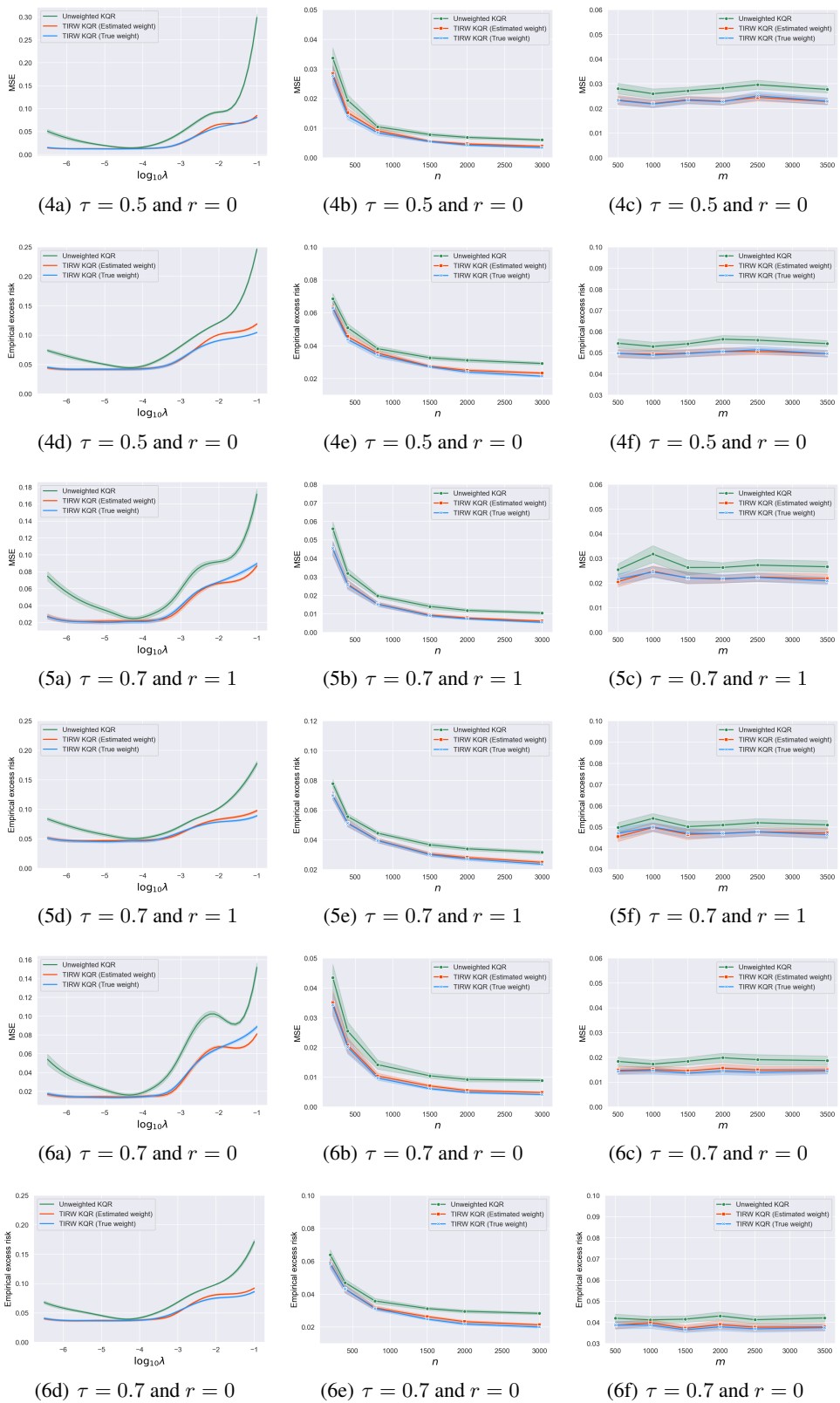

(4a) $\tau = 0.5$ and $r = 0$    (4b) $\tau = 0.5$ and $r = 0$    (4c) $\tau = 0.5$ and $r = 0$

(4d) $\tau = 0.5$ and $r = 0$    (4e) $\tau = 0.5$ and $r = 0$    (4f) $\tau = 0.5$ and $r = 0$

(5a) $\tau = 0.7$ and $r = 1$    (5b) $\tau = 0.7$ and $r = 1$    (5c) $\tau = 0.7$ and $r = 1$

(5d) $\tau = 0.7$ and $r = 1$    (5e) $\tau = 0.7$ and $r = 1$    (5f) $\tau = 0.7$ and $r = 1$

(6a) $\tau = 0.7$ and $r = 0$    (6b) $\tau = 0.7$ and $r = 0$    (6c) $\tau = 0.7$ and $r = 0$

(6d) $\tau = 0.7$ and $r = 0$    (6e) $\tau = 0.7$ and $r = 0$    (6f) $\tau = 0.7$ and $r = 0$

Figure 5: Average MSE and empirical excess risk for unweighted KQR, TIRW KQR with true weight and estimated weight, respectively (in the left panel, the curves are plotted with respect to $\log_{10} \lambda$ with $n = 500, m = 1000$; in the middle panel, the curves are plotted with respect to $n$ with fixed $m = 1000, \lambda = 10^{-4}$; in the right panel, the curves are plotted with respect to $m$ with fixed $n = 500, \lambda = 10^{-4}$)

## A.2.4 Moment bounded case in Example S4

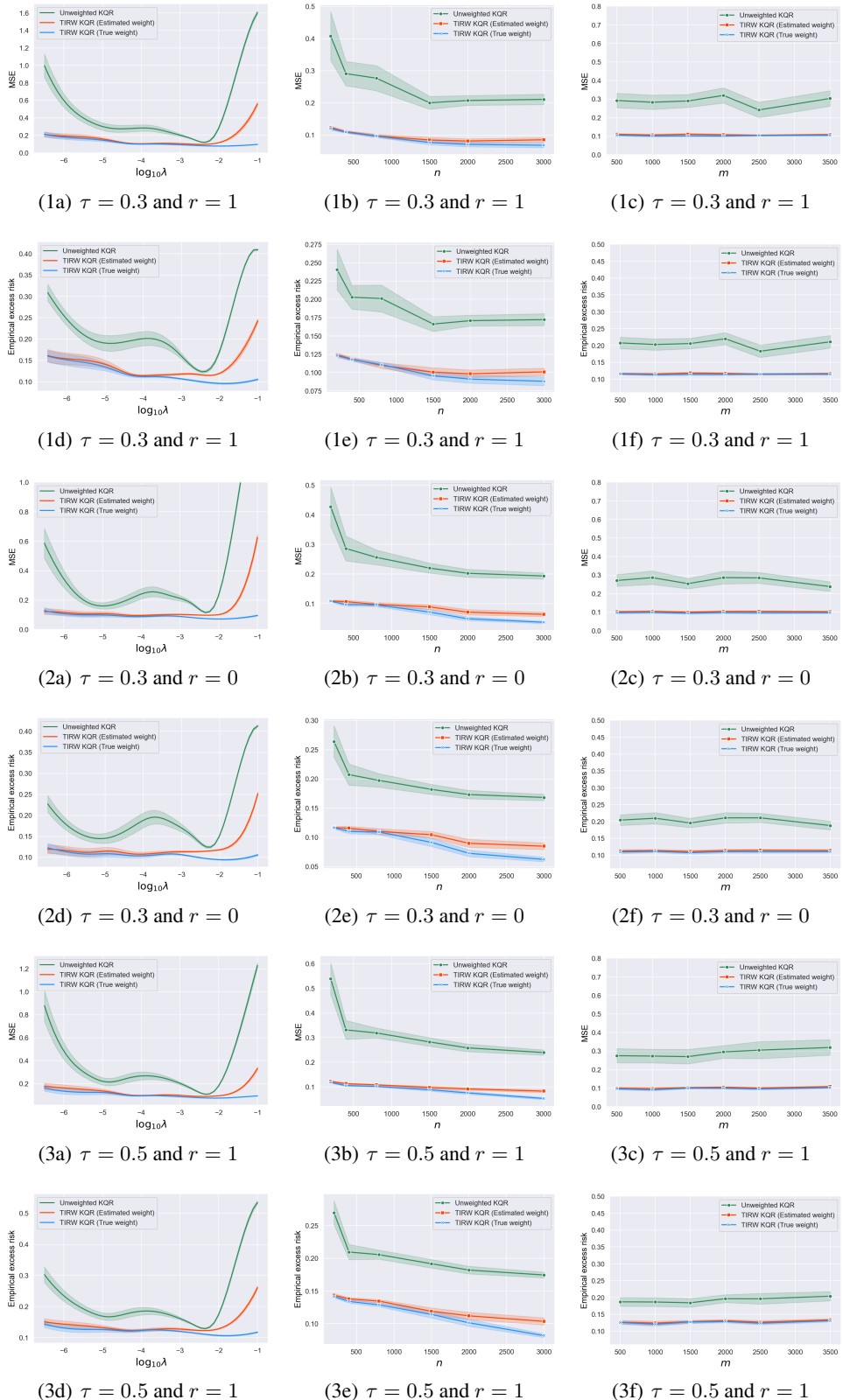

(1a) $\tau = 0.3$ and $r = 1$      (1b) $\tau = 0.3$ and $r = 1$      (1c) $\tau = 0.3$ and $r = 1$

(1d) $\tau = 0.3$ and $r = 1$      (1e) $\tau = 0.3$ and $r = 1$      (1f) $\tau = 0.3$ and $r = 1$

(2a) $\tau = 0.3$ and $r = 0$      (2b) $\tau = 0.3$ and $r = 0$      (2c) $\tau = 0.3$ and $r = 0$

(2d) $\tau = 0.3$ and $r = 0$      (2e) $\tau = 0.3$ and $r = 0$      (2f) $\tau = 0.3$ and $r = 0$

(3a) $\tau = 0.5$ and $r = 1$      (3b) $\tau = 0.5$ and $r = 1$      (3c) $\tau = 0.5$ and $r = 1$

(3d) $\tau = 0.5$ and $r = 1$      (3e) $\tau = 0.5$ and $r = 1$      (3f) $\tau = 0.5$ and $r = 1$

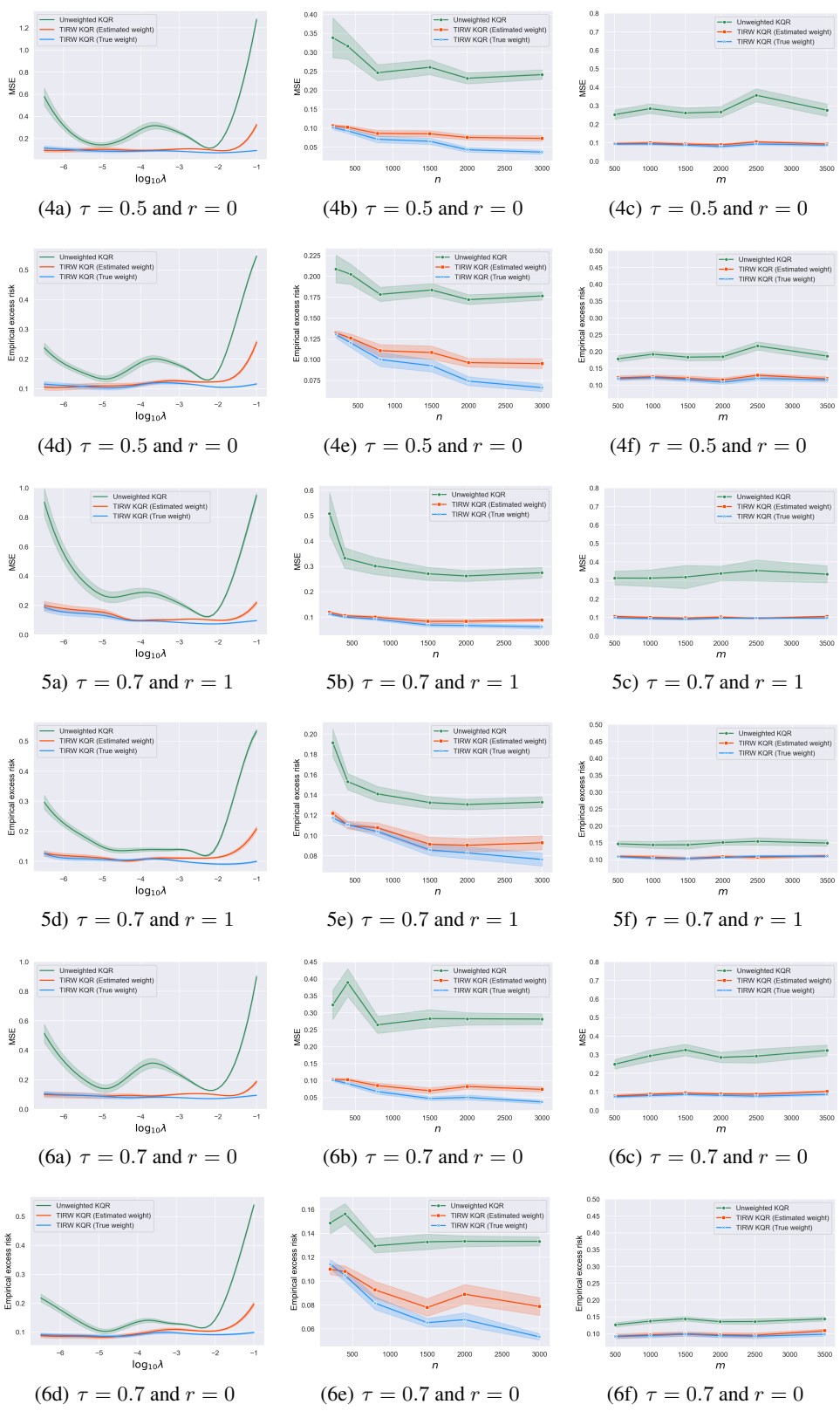

Figure 6: Average MSE and empirical excess risk for unweighted KQR, TIRW KQR with true weight and estimated weight, respectively (in the left panel, the curves are plotted with respect to $\log_{10} \lambda$ with $n = 500, m = 1000$; in the middle panel, the curves are plotted with respect to $n$ with fixed $m = 1000, \lambda = 10^{-4}$; in the right panel, the curves are plotted with respect to $m$ with fixed $n = 500, \lambda = 10^{-4}$)

### A.3 Kernel logistic regression

For the logistic loss function, we consider the following example.

**Example S5**: The response $y$ is generated by $P(y = 1 | \mathbf{x}) = \frac{1}{1+\exp(-f(\mathbf{x}))}$ where $\mathbf{x} = (x_0, x_1, x_2)^\top \in \mathcal{R}^3$ and $f(\mathbf{x}) = -x_0^2 + 3\sin(3\pi x_0) + e^{x_1^2 - x_2^2}$. We consider $\rho_{\mathbf{x}}^S(\mathbf{x}) = g(x_0; \alpha_s, \beta_s)$ and $\rho_{\mathbf{x}}^T(\mathbf{x}) = g(x_0; \alpha_t, \beta_t)$ with $\alpha_s = 2.5, \beta_s = 2, \alpha_t = 3, \beta_t = 4$ for the uniformly bounded case and $\alpha_s = 4, \beta_s = 1, \alpha_t = 3, \beta_t = 6$ for the moment bounded case, respectively. Note that for any learned $\widehat{f}$, the classification rule is specified as $\mathrm{sign}(\widehat{f}(\mathbf{x}))$. Figure 7 presents the results for the uniformly bounded case, and the results for the moment bounded case are presented in Figure 8.

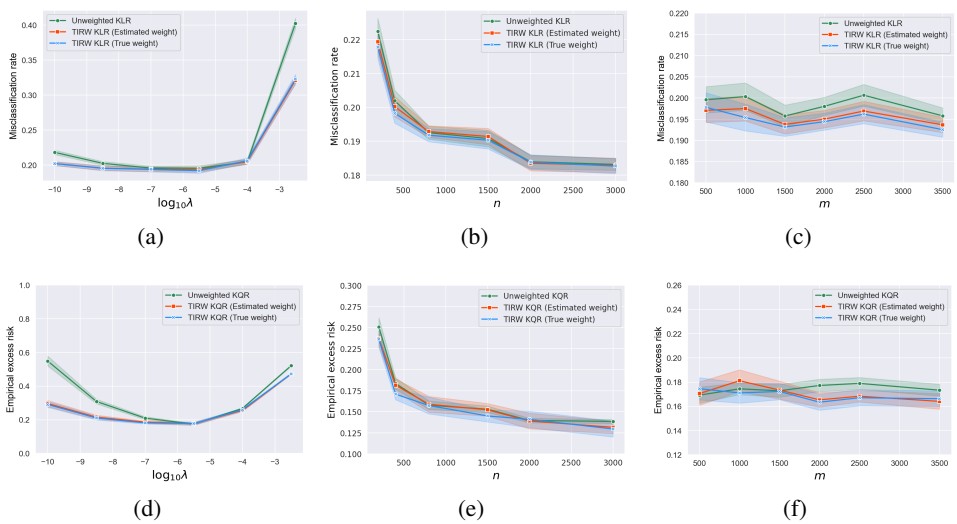

Figure 7: Average misclassification rate and empirical excess risk for unweighted KLR, TIRW KLR with true weight and estimated weight, respectively (in (a) and (d), the curves are plotted with respect to $\log_{10} \lambda$ with $n = 500, m = 1000$; in (b) and (e) the curves are plotted with respect to $n$ with fixed $m = 1000, \lambda = 5 \times 10^{-5}$; in (c) and (f), the curves are plotted with respect to $m$ with fixed $n = 500, \lambda = 5 \times 10^{-5}$)

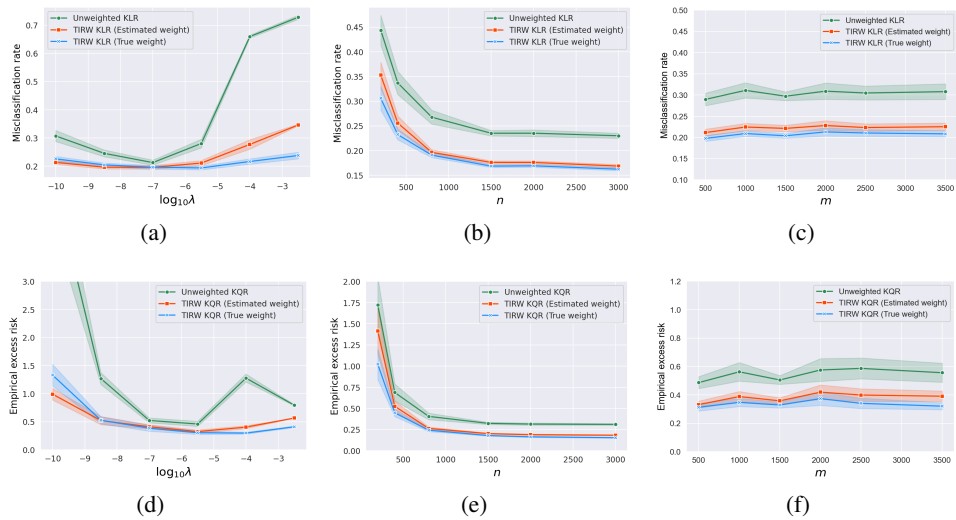

Figure 8: Average misclassification rate and empirical excess risk for unweighted KLR, TIRW KLR with true weight and estimated weight, respectively (in (a) and (d), the curves are plotted with respect to $\log_{10} \lambda$ with $n = 500, m = 1000$; in (b) and (e) the curves are plotted with respect to $n$ with fixed $m = 1000, \lambda = 5 \times 10^{-5}$; in (c) and (f), the curves are plotted with respect to $m$ with fixed $n = 500, \lambda = 5 \times 10^{-5}$)

Clearly, Figure 7 demonstrates that the difference between the performance of the TIRW estimator and the unweighted estimator is very close under the uniformly bounded case, which confirms our theoretical findings. In addition, as we found in the other examples, we can conclude from (a) and (d) in Figure 8 that the performance of the TIRW estimator, in terms of both misclassification rate and empirical excess risk, maintains relative stability for different choices of $\lambda$. But if we don't select $\lambda$ carefully, the unweighted estimator performs extremely poor. Other cases in Figure 8 confirm that the TIRW estimator takes a remarkable benefit.

## A.4 Real applications for multi-source datasets

In this section, we apply KSVM to a wide range of real datasets that are available in the UCI archive `https://archive.ics.uci.edu/ml/datasets.php`, including the ionosphere dataset, the dry bean dataset, the magic04 dataset, and the banknote authentication dataset. Specifically, the ionosphere dataset contains 350 instances and 34 covariates, and we select all the instances with 3-7-th covariates into the model. The dry bean dataset contains 13611 instances and 16 covariates, and we randomly select 30% of this dataset with the 1-8-th covariates. The magic04 dataset contains 19019 instances and 10 covariates, and we randomly select 10% of this dataset with the 2-10-th covariates. The banknote authentication dataset contains 1372 instances and 4 covariates. The numerical performance is summarized in the following table.

Table 1: Classification performance on multi-source datasets.

| Dataset | Estimator | $C = 0.01$ | $C = 0.1$ | $C = 1$ | $C = 10$ | $C = 100$ |
|---|---|---|---|---|---|---|
| Ionosphere | Unweighted | $0.260 \pm 0.007$ | $0.266 \pm 0.013$ | $0.602 \pm 0.098$ | $\mathbf{0.637 \pm 0.104}$ | $0.625 \pm 0.102$ |
| | TIRW | $0.740 \pm 0.007$ | $\mathbf{0.749 \pm 0.015}$ | $0.735 \pm 0.108$ | $0.665 \pm 0.108$ | $0.643 \pm 0.103$ |
| Dry Bean | Unweighted | $0.266 \pm 0.008$ | $0.616 \pm 0.012$ | $\mathbf{0.748 \pm 0.014}$ | $0.719 \pm 0.021$ | $0.668 \pm 0.020$ |
| | TIRW | $\mathbf{0.824 \pm 0.022}$ | $0.773 \pm 0.013$ | $0.764 \pm 0.014$ | $0.713 \pm 0.022$ | $0.696 \pm 0.028$ |
| Magic04 | Unweighted | $0.621 \pm 0.006$ | $0.625 \pm 0.007$ | $\mathbf{0.779 \pm 0.011}$ | $0.752 \pm 0.012$ | $0.744 \pm 0.012$ |
| | TIRW | $0.621 \pm 0.006$ | $0.624 \pm 0.007$ | $\mathbf{0.804 \pm 0.041}$ | $0.768 \pm 0.011$ | $0.748 \pm 0.013$ |
| Authentication | Unweighted | $0.237 \pm 0.007$ | $0.718 \pm 0.031$ | $\mathbf{0.940 \pm 0.033}$ | $0.920 \pm 0.036$ | $0.919 \pm 0.036$ |
| | TIRW | $0.811 \pm 0.037$ | $\mathbf{0.986 \pm 0.015}$ | $0.973 \pm 0.023$ | $0.928 \pm 0.032$ | $0.919 \pm 0.036$ |

As shown in Table 1, the TIRW estimator outperforms the unweighted estimator on each dataset for almost all the choices of $C_\lambda$. We also observe that the unweighted estimator has a much less satisfying accuracy of prediction for small choices of $C_\lambda$. Nevertheless, with importance ratio correction, the accuracy rate has been significantly improved for small choices of $C_\lambda$, even attaining nearly optimal for the first two datasets. For a large choice of $C_\lambda$, these two estimators have a negligible gap in accuracy rate.

## A.5 Kullback-Leibler importance estimation procedure

In this section, we introduce the importance ratio estimation procedure based on Kullback-Leibler divergence (Sugiyama et al., 2007b). Recall that we have the source input data $\mathbf{x}_1^S, ..., \mathbf{x}_n^S$ generated from $\rho_{\mathbf{x}}^S$ and the target input data $\mathbf{x}_1^T, ..., \mathbf{x}_m^T$ generated from $\rho_{\mathbf{x}}^T$ and our goal is to estimate the ratio $\phi(\mathbf{x})$. Since $\rho_{\mathbf{x}}^T(\mathbf{x}) = \phi(\mathbf{x})\rho_{\mathbf{x}}^S(\mathbf{x})$, the true ratio $\phi(\mathbf{x})$ can be correctly identified by solving the population version of the optimization task that

$$\underset{g(\mathbf{x})}{\text{minimize}} \quad \mathrm{KL}\big(\rho_{\mathbf{x}}^T(\mathbf{x}) \| g(\mathbf{x})\rho_{\mathbf{x}}^S(\mathbf{x})\big), \tag{1}$$

where $\mathrm{KL}(p\|q) = \int p(\mathbf{x}) \log \frac{p(\mathbf{x})}{q(\mathbf{x})} d\mathbf{x}$ denotes Kullback–Leibler divergence between $P$ and $Q$ with probability densities $p$ and $q$ respectively. Note that either the bounded case or second moment bounded case considered in the main text, $\rho_{\mathbf{x}}^T(\mathbf{x})$ is absolutely continuous with respect to $\rho_{\mathbf{x}}^S(\mathbf{x})$, which ensures the optimization problem (1) is well defined. Since

$$\mathrm{KL}(\rho_{\mathbf{x}}^T(\mathbf{x})\|g(\mathbf{x})\rho_{\mathbf{x}}^S(\mathbf{x})) = \int \rho_{\mathbf{x}}^T(\mathbf{x}) \log \left(\frac{\rho_{\mathbf{x}}^T(\mathbf{x})}{g(\mathbf{x})\rho_{\mathbf{x}}^S(\mathbf{x})}\right) d\mathbf{x}$$

$$= \int \rho_{\mathbf{x}}^T(\mathbf{x}) \log \left(\frac{\rho_{\mathbf{x}}^T(\mathbf{x})}{\rho_{\mathbf{x}}^S(\mathbf{x})}\right) d\mathbf{x} - \int \rho_{\mathbf{x}}^T(\mathbf{x}) \log g(\mathbf{x}) d\mathbf{x},$$

we can rewrite the objective function as $-\int \rho_{\mathbf{x}}^T(\mathbf{x})\log(g(\mathbf{x}))d\mathbf{x}$ by ignoring the constant term. It can be approximated by its empirical version that $-\frac{1}{m}\sum_{i=1}^m \log g(\mathbf{x}_i^T)$ with $g(\mathbf{x}) = \sum_{k=1}^b \alpha_k K(\mathbf{x}_k, \mathbf{x})$ and $\{\mathbf{x}_1, ..., \mathbf{x}_b\}$ denoting a subset of target input data with $b$ as a pre-fixed number and $K(\cdot, \cdot)$ denoting some certain kernel function. Since the true ratio $\phi(\mathbf{x})$ is non-negative and satisfies $\frac{1}{n}\sum_{i=1}^n \phi(\mathbf{x}_i^S) \approx \int \phi(\mathbf{x})\rho_{\mathbf{x}}^S(\mathbf{x})d\mathbf{x} = 1$. We add some constraints that $\alpha_k \geq 0$ and $\frac{1}{m}\sum_{i=1}^n \sum_{k=1}^b \alpha_k K(\mathbf{x}_k, \mathbf{x}_i^S) = 1$. This leads to the optimization problem:

$$\underset{\boldsymbol{\alpha} \in \mathcal{R}^b}{\text{maximum}} \quad \frac{1}{m}\sum_{i=1}^m \log\left(\sum_{k=1}^b \alpha_k K(\mathbf{x}_k, \mathbf{x}_i^T)\right),$$
$$\text{s.t.} \quad \alpha_k \geq 0, \; k = 1, ..., b,$$
$$\frac{1}{m}\sum_{i=1}^n \sum_{k=1}^b \alpha_k K(\mathbf{x}_k, \mathbf{x}_i^S) = 1.$$

### A.6   Importance weighted cross validation

Sugiyama et al. (2007a) points out that cross-validation (CV) on the unweighted training data introduces an additional source of bias in making predictions on test data due to covariate shift. They propose a method called importance weighted cross validation (IWCV) according to important ratio to compensate for the effect of covariate shift. First, one can randomly divide the training set $\{(\mathbf{x}_i, y_i)\}_{i=1}^n$ into $b$ disjoint non-empty subsets $\{\mathcal{T}_i\}_{k=1}^b$. Then, denoting the learned function by using dataset $\{\mathcal{T}_k\}_{k \neq j}$ as $\widehat{f}_j$. Instead of the classical CV procedure, the IWCV aims to minimize

$$\widehat{R}_{IWCV} = \frac{1}{b}\sum_{k=1}^b \frac{1}{|\mathcal{T}_k|}\sum_{(\mathbf{x}_i, y_i) \in \mathcal{T}_k} \phi(\mathbf{x}_i)L\big(y_i, \widehat{f}_k(\mathbf{x}_i)\big).$$

### A.7   Discussion on Assumption 2

Assumption 2 in the main text is a local $c_0$-strongly convexity condition on the expected loss function with respect to $\mathcal{L}^2(\mathcal{X}, P_{\mathbf{x}}^S)$ and $\mathcal{L}^2(\mathcal{X}, P_{\mathbf{x}}^T)$ at $f^*$. So verifying Assumption 2 is equivalent to verifying the local $c_0$ strongly convexity of the loss function. Here are some examples:

- For the squared loss $L(y, f(\mathbf{x})) = (y - f(\mathbf{x}))^2$, note that for any $y \in \mathcal{R}$, the function $z \to (y - z)^2$ is strongly convex with parameter $c_0 = 1$, so $f \to L(y, f(\mathbf{x}))$ satisfies the condition in Assumption 2 with $c_0 = 1$.

- For the Huber loss $L(y, f(\mathbf{x})) = (y - f(\mathbf{x}))^2$, if $|y - f(\mathbf{x})| \leq \delta$; $\delta|y - f(\mathbf{x})| - \frac{1}{2}\delta^2$, otherwise, since this loss function is locally equivalent to the squared loss, so it is locally strongly convex under mild tail condition on $y - f^*(\mathbf{x})$.

- For the check loss $L(y, f(\mathbf{x})) = (y - f(\mathbf{x}))\big(\tau - I_{\{y \leq f(\mathbf{x})\}}\big)$, the local strong convexity holds if the conditional density of $y - f^*(\mathbf{x})$ given $\mathbf{x}$ is bounded away from $c_0$ uniformly (Lian, 2022).

For the other loss functions, including logistic loss and hinge loss, more detailed discussions and verifications can be found on Pages 470-472 in Wainwright (2019). Interested readers are referred to it for more details.

## B   Algorithm details

In this section, we provide the computing details for different loss functions considered in our experiments.

**Kernel ridge regression.** For the squared loss, the minimizer $\widehat{f}$ takes the form of $\widehat{f}(\mathbf{x}) = \sum_{i=1}^n \widehat{\alpha}_i K(\mathbf{x}_i, \mathbf{x})$, due to the representer theorem (Smale & Zhou, 2007). Let $\widehat{\boldsymbol{\alpha}} = (\widehat{\alpha}_1, ..., \widehat{\alpha}_n)^\top \in \mathcal{R}^n$, then the solution is given by $\widehat{\boldsymbol{\alpha}} = (\mathbf{K}W\mathbf{K} + n\lambda\mathbf{K})^{-1}\mathbf{K}\,\mathbf{W}\,\mathbf{y} = (\mathbf{W}\,\mathbf{K} + n\lambda I)^{-1}\mathbf{W}\,\mathbf{y}$. Here

$\mathbf{K}$ is $n \times n$ invertible matrix with elements $K(\mathbf{x}_i, \mathbf{x}_j)$, $\mathbf{W}$ is a diagonal matrix with $W_{ii} = \widehat{\phi}_n(\mathbf{x}_i)$ and $\mathbf{y} = (y_1, ..., y_n)^\top$, where $\widehat{\phi}_n(\mathbf{x})$ is the truncation version of the KLIEP estimator $\widehat{\phi}(\mathbf{x})$.

**Kernel quantile regression.** For the check loss, there is no explicit form for the solution. Then, we attempt to search the optimal solution based on the solution of the dual problem (Takeuchi et al., 2006). In the presence of covariate shift, we derive the dual problem to be the following convex optimization task (Boyd & Vandenberghe, 2004) that

$$\text{minimize} \quad \frac{1}{2}\boldsymbol{\alpha}^\top \mathbf{K}\boldsymbol{\alpha} - \mathbf{y}^\top \boldsymbol{\alpha},$$

$$\text{s.t.} \quad C_\lambda(\tau - 1)\widehat{\phi}_n(\mathbf{x}_i) \leq \alpha_i \leq C_\lambda \tau \widehat{\phi}_n(\mathbf{x}_i), \text{ for } 1 \leq i \leq n,$$

$$\vec{1}^\top \boldsymbol{\alpha} = 0,$$

where $\vec{1}$ denotes the vector whose elements are 1, $\mathbf{y} = (y_1, ..., y_n)^\top$ and $C_\lambda = 1/(n\lambda)$. $b$ is the dual variable to the constraint $\vec{1}^\top \boldsymbol{\alpha} = 0$.

**Kernel support vector machine.** For the hinge loss, we also solve the duality problem (Schölkopf et al., 2002). Specifically, in the presence of covariate shift, we solve the following convex optimization task that

$$\text{minimize} \quad \frac{1}{2}\boldsymbol{\eta}^\top \widetilde{\mathbf{K}}\boldsymbol{\eta} - \vec{1}^\top \boldsymbol{\eta},$$

$$\text{s.t.} \quad \mathbf{y}^\top \boldsymbol{\eta} = 0,$$

$$0 \leq \eta_i \leq C_\lambda \widehat{\phi}_n(\mathbf{x}_i), \quad i = 1, \cdots, n,$$

where $\widetilde{\mathbf{K}}$ denotes the $n \times n$ matrix with entries $K(x_i, x_j)y_i y_j$ and $\boldsymbol{\alpha} = (\eta_1 y_1, ..., \eta_n y_n)^\top$. $b$ is the dual variable to the constraint $\mathbf{y}^\top \boldsymbol{\eta} = 0$.

**Kernel logistic regression.** For the logistic loss, we use the Newton–Raphson algorithm to solve the optimization task (Keerthi et al., 2005). Let $\mathbf{K}_i$ correspond the $i$-th row of $\mathbf{K}$ and $\widehat{\boldsymbol{\phi}} = (\widehat{\phi}(\mathbf{x}_1), ..., \widehat{\phi}(\mathbf{x}_n))^\top$, $\boldsymbol{p} = (p_1, ..., p_n)^\top$ with $p_i = \frac{\exp(\mathbf{K}_i \boldsymbol{\alpha})}{1 + \exp(\mathbf{K}_i \boldsymbol{\alpha})}$. We conduct the iterative algorithm

$$\boldsymbol{\alpha}_{k+1} = \boldsymbol{\alpha}_k - J(\boldsymbol{\alpha}_k)^{-1}F(\boldsymbol{\alpha}_k) = (\mathbf{K}\mathbf{W}\mathbf{K} + \lambda\mathbf{K})^{-1}(\mathbf{K}\mathbf{W}\mathbf{K}\boldsymbol{\alpha}_k + \mathbf{K}\mathbf{y}\odot(1-\boldsymbol{p})\odot\widehat{\boldsymbol{\phi}})$$

$$= (\mathbf{W}\mathbf{K} + \lambda I)^{-1}(\mathbf{W}\mathbf{K}\boldsymbol{\alpha}_k + \mathbf{y}\odot(1-\boldsymbol{p})\odot\widehat{\boldsymbol{\phi}}),$$

where $W$ denotes the diagonal matrix with $W_{ii} = \widehat{\phi}_n(\mathbf{x}_i)p_i(1-p_i)$ and $\mathbf{a}\odot\mathbf{c} = (a_1 c_1, ..., a_n c_n)^\top$ for two vectors $\mathbf{a}$ and $\mathbf{c}$.

## C Technical proofs

This part provides the proofs of all the theorems and corollaries in the main text. Note that our theoretical analysis mainly employs the symmetrization technique and concentration inequality in learning theory. For the second moment bounded case, Section C.1 is devoted to the proof of Theorem 3 that indicates the TIRW estimator achieves optimal rate, and Section C.2 gives the proof of Theorem 2 showing the sub-optimal rate for the unweighted estimator. For the uniformly bounded case, Section C.3 presents the proof of Theorem 1 to show the optimal rate for the unweighted estimator, which follows a similar argument as in Section C.2. Section C.4 gives the detailed derivation of all the corollaries. Section C.5 discusses the results of the minimax lower bound when some specified loss functions are used. In Section C.6, we further discuss the theoretical gap with the importance ratio replaced by its plugin estimator and potential future direction. For ease of notation, we discard the superscripts of $\mathbf{x}_i^S$ and $y_i^S$ to $\mathbf{x}_i$ and $y_i$ in our proofs, that is, $\{\mathbf{x}_i, y_i\}_{i=1}^n$ is driven from the source model. Additionally, we define $P\varphi := E_S[\varphi(\mathbf{x}, y)]$ and $P_n\varphi := (1/n)\sum_{i=1}^n \varphi(\mathbf{x}_i, y_i)$ for a measurable function $\varphi(\mathbf{x}, y)$, and clarify that the expectation $E[\cdot]$ in our proof is taking with respect to all random variables contained in it. Note that we remain $\|f^*\|_K$ in our proof and the theoretical results in the main text can be obtained by letting $\|f^*\|_K = 1$.

### C.1 Proof of Theorem 3

The following Lemma states Talagrand's concentration inequality for random elements taking values in some space $\mathcal{Z}$. One can refer to Bousquet (2002) for detailed proof.

**Lemma C.1.1.** *Let $Z_1, \ldots, Z_n$ be independent random elements taking values in some space $\mathcal{Z}$ and let $\Xi$ be a class of real-valued functions on $\mathcal{Z}$, if we have*

$$\|\xi\| \leq \eta_n \quad and \quad \frac{1}{n} \sum_{i=1}^{n} \text{Var}\left(\xi\left(Z_i\right)\right) \leq \zeta_n^2, \quad \forall \xi \in \Xi.$$

*Define $\boldsymbol{Z} := \sup_{\xi \in \Xi} \left|\frac{1}{n} \sum_{i=1}^{n} \left(\xi\left(Z_i\right) - E\xi\left(Z_i\right)\right)\right|$. Then for $t > 0$*

$$P\left(\boldsymbol{Z} \geq E(\boldsymbol{Z}) + t\sqrt{2\left(\zeta_n^2 + 2\eta_n E(\boldsymbol{Z})\right)} + \frac{2\eta_n t^2}{3}\right) \leq \exp\left(-nt^2\right).$$

The following Lemma is the core of our proofs. It bounds the supremum of the difference between the empirical average dependent on the source data and the target expectation within a local ball using the Rademacher complexity function and Lemma C.1.1.

**Lemma C.1.2.** *For any radii $\delta > 0$, we define event $\mathcal{K}(\delta)$ as*

$$\sup_{f \in \Theta(\delta)} \left|\frac{1}{n} \sum_{i=1}^{n} \phi_n(\mathbf{x}_i)\left(L(y_i, f(\mathbf{x}_i)) - L(y_i, f^*(\mathbf{x}_i))\right) - E_T\left[L(y, f(\mathbf{x})) - L(y, f^*(\mathbf{x}))\right]\right| \leq \mathcal{M}(\delta),$$

*where $\Theta(\delta) := \{f \in \mathcal{H}_K \mid \|f - f^*\|_T \leq \delta, and \|f - f^*\|_K \leq 3\|f^*\|_K\}$ and $\mathcal{M}(\delta) = C\sqrt{\beta^2 \log n} R(\delta)$, then $\mathcal{K}(\delta)$ holds with probability at least $1 - n^{-c_3}$.*

Before providing the detailed proof of Lemma C.1.2, we give some discussions to illustrate the motivation of the proof. Specifically, Lemma C.1.2 states a general uniform law for the Lipschitz loss functions under covariate shift. Note that by the empirical process theory, the empirical average $\frac{1}{n} \sum_{i=1}^{n} \phi_n(\mathbf{x}_i)\left(L(y_i, f(\mathbf{x}_i)) - L(y_i, f^*(\mathbf{x}_i))\right)$ approximates its population counterpart $E_S[\phi_n(\mathbf{x})(L(y, f(\mathbf{x})) - L(y, f^*(\mathbf{x})))]$ uniformly on some function class. When the truncation $\gamma_n$ diverges fast enough as $n$ grows, the quantity $|E_S[\phi_n(\mathbf{x})(L(y, f(\mathbf{x})) - L(y, f^*(\mathbf{x})))] - E_T[L(y, f(\mathbf{x})) - L(y, f^*(\mathbf{x}))]|$ is negligible. Therefore, it is expected that $\frac{1}{n} \sum_{i=1}^{n} \phi_n(\mathbf{x}_i)\left(L(y_i, f(\mathbf{x}_i)) - L(y_i, f^*(\mathbf{x}_i))\right)$ is close to $E_T[L(y, f(\mathbf{x})) - L(y, f^*(\mathbf{x}))]$ uniformly. Our proof is precisely motivated by this intuition. Moreover, in the proof, we decompose the total error into the empirical error $\sup_{f \in \Theta(\delta)} |\frac{1}{n} \sum_{i=1}^{n} \phi_n(\mathbf{x}_i)\left(L(y_i, f(\mathbf{x}_i)) - L(y_i, f^*(\mathbf{x}_i))\right) - E_S[\phi_n(\mathbf{x})(L(y, f(\mathbf{x})) - L(y, f^*(\mathbf{x})))]|$ and the approximation error $\sup_{f \in \Theta(\delta)} |E_S[\phi_n(\mathbf{x})(L(y, f(\mathbf{x})) - L(y, f^*(\mathbf{x})))] - E_T[L(y, f(\mathbf{x})) - L(y, f^*(\mathbf{x}))]|$. Recall that $\phi_n(\mathbf{x}) = \min\{\phi(\mathbf{x}), \gamma_n\}$, and here, the truncation parameter $\gamma_n$ plays a key role in balancing such two errors. For example, a fast diverging $\gamma_n$ may reduce the approximation error but compromise the empirical error. Therefore, an optimal $\gamma_n$ is the one that diverges at a certain rate (i.e., $\gamma_n = O(\sqrt{n})$) to achieve the optimal tradeoff between empirical and approximation errors.

**Proof of Lemma C.1.2.** We first make the following decomposition that

$$\sup_{f \in \Theta(\delta)} \left|\frac{1}{n} \sum_{i=1}^{n} \phi_n(\mathbf{x}_i)\left(L(y_i, f(\mathbf{x}_i)) - L(y_i, f^*(\mathbf{x}_i))\right) - E_T\left[L(y, f(\mathbf{x})) - L(y, f^*(\mathbf{x}))\right]\right|$$

$$\leq \underbrace{\sup_{f \in \Theta(\delta)} \left|\frac{1}{n} \sum_{i=1}^{n} \phi_n(\mathbf{x}_i)\left(L(y_i, f(\mathbf{x}_i)) - L(y_i, f^*(\mathbf{x}_i))\right) - E_S\left[\phi_n(\mathbf{x})\left(L(y, f(\mathbf{x})) - L(y, f^*(\mathbf{x}))\right)\right]\right|}_{\text{Empirical error } D_1}$$

$$+ \underbrace{\sup_{f \in \Theta(\delta)} \left|E_S\left[\phi_n(\mathbf{x})\left(L(y, f(\mathbf{x})) - L(y, f^*(\mathbf{x}))\right)\right] - E_T\left[L(y, f(\mathbf{x})) - L(y, f^*(\mathbf{x}))\right]\right|}_{\text{Approximation error } D_2}.$$

Then, we only need to bound $D_1$ and $D_2$ separately. To bound $D_1$, we firstly use the standard symmetrization technique in empirical process (Pollard, 2012; Wainwright, 2019) to bound $E[D_1]$

that

$$E[D_1] = E\Big[ \sup_{f \in \Theta(\delta)} \big| (P_n - P)\phi_n(\mathbf{x}) \left( L(y, f(\mathbf{x})) - L(y, f^*(\mathbf{x})) \right) \big| \Big]$$

$$\overset{(i)}{\leq} \frac{2}{n} E\Big[ \sup_{f \in \Theta(\delta)} \big| \sum_{i=1}^n \sigma_i \phi_n(\mathbf{x}_i) \big( L(y_i, f(\mathbf{x}_i)) - L(y_i, f^*(\mathbf{x}_i)) \big) \big| \Big] \tag{2}$$

$$\overset{(ii)}{\leq} \frac{4c_L}{n} E\Big[ \sup_{f \in \Theta(\delta)} \big| \sum_{i=1}^n \sigma_i \phi_n(\mathbf{x}_i)(f(\mathbf{x}_i) - f^*(\mathbf{x}_i)) \big| \Big],$$

where $\{\sigma_i\}'s$ denote the Rademacher variables taking values in $\{-1, 1\}$ with equal probability, the inequality (i) is from the symmetrization technique that for any class of measurable function $\mathcal{F}$, we have $E[\sup_{\varphi \in \mathcal{F}}(P_n - P)\varphi] \leq 2E[\sup_{\varphi \in \mathcal{F}}(1/n)\sum_{i=1}^n \sigma_i \varphi(\mathbf{x}_i, y_i)]$. The inequality (ii) follows from the fact that the loss function is $c_L$-Lipschitz continuous and the Ledoux–Talagrand contraction inequality (Wainwright, 2019).

For any $f \in \Theta(\delta)$, we denote $g = f - f^* \in \mathcal{H}_K$ and then, there holds $g = \sum_{j=1}^\infty g_j \psi_j$ with $g_j = \int_{\mathcal{X}} f(\mathbf{x})\psi_j(\mathbf{x})\rho_{\mathbf{x}}^T(\mathbf{x})d\mathbf{x}$. Clearly, we have $\|g\|_T \leq \delta$ and $\|g\|_K \leq 3\|f^*\|_K$, which implies that $\sum_{j=1}^\infty g_j^2 \leq \delta^2$ and $\sum_{j=1}^\infty g_j^2/\mu_j \leq 9\|f^*\|_K^2$. Combining these two results, there holds

$$\sum_{j=1}^\infty \frac{g_j^2}{\min(\delta^2, \mu_j\|f^*\|_K^2)} \leq 10. \tag{3}$$

Then, we have

$$\Big| \sum_{i=1}^n \sigma_i \phi_n(\mathbf{x}_i)(f(\mathbf{x}_i) - f^*(\mathbf{x}_i)) \Big| = \Big| \sum_{i=1}^n \sigma_i \phi_n(\mathbf{x}_i) \sum_{j=1}^\infty g_j \psi_j(\mathbf{x}_i) \Big|$$

$$= \Big| \sum_{j=1}^\infty \frac{g_j}{\sqrt{\min(\delta^2, \mu_j\|f^*\|_K^2)}} \sqrt{\min(\delta^2, \mu_j\|f^*\|_K^2)} \sum_{i=1}^n \sigma_i \phi_n(\mathbf{x}_i)\psi_j(\mathbf{x}_i) \Big| \tag{4}$$

$$\overset{(i)}{\leq} \sqrt{10} \left\{ \sum_{j=1}^\infty \min(\delta^2, \mu_j\|f^*\|_K^2) \left( \sum_{i=1}^n \sigma_i \phi_n(\mathbf{x}_i)\psi_j(\mathbf{x}_i) \right)^2 \right\}^{1/2},$$

where the inequality (i) follows from Cauthy-Schwarz inequality and the fact (3). Moreover, by plugging (4) into (2), we have

$$E[D_1] \leq \frac{4\sqrt{10}c_L}{n} E\Big[ \sum_{j=1}^\infty \min(\delta^2, \mu_j\|f^*\|_K^2) \left( \sum_{i=1}^n \sigma_i \phi_n(\mathbf{x}_i)\psi_j(\mathbf{x}_i) \right)^2 \Big]^{1/2}$$

$$\overset{(i)}{\leq} \frac{4\sqrt{10}c_L}{n} \left\{ \sum_{j=1}^\infty \min(\delta^2, \mu_j\|f^*\|_K^2) E_{\mathbf{x}, \sigma}\left[ \sum_{i=1}^n \sigma_i \phi_n(\mathbf{x}_i)\psi_j(\mathbf{x}_i) \right]^2 \right\}^{1/2}$$

$$\overset{(ii)}{=} \frac{4\sqrt{10}c_L}{n} \left\{ \sum_{j=1}^\infty \min(\delta^2, \mu_j\|f^*\|_K^2) \sum_{i=1}^n E_{\mathbf{x}, \sigma}\left[ \sigma_i^2 \phi_n^2(\mathbf{x}_i)\psi_j^2(\mathbf{x}_i) \right] \right\}^{1/2}$$

$$\overset{(iii)}{\leq} \frac{4\sqrt{10}c_L}{n} \left\{ \sum_{j=1}^\infty \min(\delta^2, \mu_j\|f^*\|_K^2) \sum_{i=1}^n E[\phi^2(\mathbf{x}_i)] \right\}^{1/2},$$

where the first inequality (i) follows from Jensen's inequality, the second inequality (ii) follows from the fact that $E_{\mathbf{x}, \sigma}[\sigma_i \phi_n(\mathbf{x}_i)\psi_j(\mathbf{x}_i)] = 0$ for each $i$, and the last inequality (iii) follows from the assumption that $\|\psi_j\|_\infty \leq 1$ for all $j \geq 1$ and the fact that $\phi_n(\mathbf{x}_i) \leq \phi(\mathbf{x}_i)$. Note that $E[\phi^2(\mathbf{x}_i)] \leq \beta^2$, and thus we have

$$E(D_1) \leq 4\sqrt{10} \sqrt{\frac{\beta^2 c_L^2}{n} \sum_{j=1}^\infty \min(\delta^2, \mu_j\|f^*\|_K^2)} = 4\sqrt{10\beta^2} c_L R(\delta). \tag{5}$$

Next, we turn to bound $D_1 - E(D_1)$. Recall that $\sum_{j=1}^{\infty} \frac{g_j^2}{\min(\delta^2, \mu_j \|f^*\|_K^2)} \leq 10$ and $\|\psi_j\|_{\infty} \leq 1$, and then there holds

$$|g(\mathbf{x})| = \left| \sum_{j=1}^{\infty} g_j \psi_j(\mathbf{x}) \right| \overset{(i)}{\leq} \sqrt{\sum_{j=1}^{\infty} \frac{g_j^2}{\min(\delta^2, \mu_j \|f^*\|_K^2)}} \sqrt{\sum_{j=1}^{\infty} \min(\delta^2, \mu_j \|f^*\|_K^2) \psi_j^2(\mathbf{x})}$$

$$\leq \sqrt{10 \sum_{j=1}^{\infty} \min(\delta^2, \mu_j \|f^*\|_K^2)} = \sqrt{10n} R(\delta),$$

where the first inequality (i) follows from Cauthy-Schwarz inequality. Consequently, we have

$$|\phi_n(\mathbf{x}_i)\left(L(y_i, f(\mathbf{x}_i)) - L(y_i, f^*(\mathbf{x}_i))\right)| \leq \gamma_n \left|(L(y_i, f(\mathbf{x}_i)) - L(y_i, f^*(\mathbf{x}_i)))\right|$$

$$\leq \gamma_n c_L |g(\mathbf{x}_i)| \leq \gamma_n c_L \sqrt{10n} R(\delta) = \sqrt{10\beta^2} n c_L R(\delta),$$

where we use $\gamma_n = \sqrt{n\beta^2}$. Furthermore, we have

$$E\left[\left\{\phi_n(\mathbf{x}_i)\left(L(y_i, f(\mathbf{x}_i)) - L(y_i, f^*(\mathbf{x}_i))\right)\right\}^2\right] \leq c_L^2 E\left[\phi_n^2(\mathbf{x}_i) g^2(\mathbf{x}_i)\right] \leq 10n\beta^2 c_L^2 R^2(\delta),$$

where we use the fact that $E[\phi_n^2(\mathbf{x}_i)] \leq E[\phi^2(\mathbf{x}_i)] \leq \beta^2$. Clearly, all the required conditions in Lemma C.1.1 are satisfied by taking $\eta_n = \sqrt{10\beta^2} n c_L R(\delta)$ and $\zeta_n^2 = 10n\beta^2 c_L^2 R^2(\delta)$. Let $t = \sqrt{\frac{c_3 \log n}{n}}$, with probability at least $1 - n^{-c_3}$, there holds that

$$D_1 - E[D_1] \leq \sqrt{\frac{c_3 \log n}{n}\left(20n\beta^2 c_L^2 R^2(\delta) + 4n c_L \sqrt{10\beta^2} R(\delta) E[D_1]\right)} + \frac{2\sqrt{10\beta^2} c_3 c_L}{3} \log n R(\delta)$$

$$\overset{(i)}{\leq} 3c_L \sqrt{20 c_3 \beta^2 \log n} R(\delta) + \frac{2\sqrt{10\beta^2} c_3 c_L}{3} \log n R(\delta)$$

$$\leq \left(3c_L \sqrt{20c_3} + \frac{2\sqrt{10} c_3 c_L}{3}\right) \sqrt{\beta^2} \log n R(\delta),$$

(6)

where the first inequality (i) follows from (5).

Then, combining (5) and (6), with probability at least $1 - n^{-c_3}$, we have

$$D_1 \leq C\sqrt{\beta^2} \log n R(\delta),$$

(7)

where $C = 4\sqrt{10} c_L + 3c_L \sqrt{20c_3} + \frac{2\sqrt{10\beta^2} c_3 c_L}{3}$.

Now we turn to bound $D_2$. Note that

$$D_2 \leq \sup_{f \in \Theta(\delta)} \left|E_T\left[L(y, f(\mathbf{x})) - L(y, f^*(\mathbf{x}))\right] - E_T\left[I_{\{\phi(\mathbf{x}) \leq \gamma_n\}}\left(L(y, f(\mathbf{x})) - L(y, f^*(\mathbf{x}))\right)\right]\right| +$$

$$\gamma_n \sup_{f \in \Theta(\delta)} \left|E_S\left[I_{\{\phi(\mathbf{x}) > \gamma_n\}}\left(L(y, f(\mathbf{x})) - L(y, f^*(\mathbf{x}))\right)\right]\right|$$

$$= \sup_{f \in \Theta(\delta)} \left|E_T\left[I_{\{\phi(\mathbf{x}) > \gamma_n\}}\left(L(y, f(\mathbf{x})) - L(y, f^*(\mathbf{x}))\right)\right]\right| +$$

$$\gamma_n \sup_{f \in \Theta(\delta)} \left|E_S\left[I_{\{\phi(\mathbf{x}) > \gamma_n\}}\left(L(y, f(\mathbf{x})) - L(y, f^*(\mathbf{x}))\right)\right]\right|$$

$$\leq E_T\left[I_{\{\phi(\mathbf{x}) > \gamma_n\}} \sup_{f \in \Theta(\delta)} |L(y, f(\mathbf{x})) - L(y, f^*(\mathbf{x}))|\right] +$$

$$\gamma_n E_S\left[I_{\{(\phi(\mathbf{x}) > \gamma_n\}} \sup_{f \in \Theta(\delta)} |L(y, f(\mathbf{x})) - L(y, f^*(\mathbf{x}))|\right]$$

$$\leq c_L E_T\left[I_{\{\phi(\mathbf{x}) > \gamma_n\}}\right] \sup_g \|g\|_{\infty} + \gamma_n c_L E_S\left[I_{\{\phi(\mathbf{x}) > \gamma_n\}}\right] \sup_g \|g\|_{\infty}$$

$$\overset{(i)}{\leq} \frac{\beta^2 c_L}{\gamma_n} \sqrt{10n} R(\delta) + \gamma_n \frac{\beta^2 c_L}{\gamma_n^2} \sqrt{10n} R(\delta) \leq 2\sqrt{10\beta^2} c_L R(\delta),$$

(8)

where the inequality (i) follows from Markov inequality.

Combining (7) and (8), with probability at least $1 - n^{-c_3}$, there holds

$$D_1 + D_2 \le C\sqrt{\beta^2}\log nR(\delta) = \mathcal{M}(\delta),$$

where $C = 4\sqrt{10}c_L + 3c_L\sqrt{20c_3} + \frac{2\sqrt{10\beta^2 c_3 c_L}}{3} + 2\sqrt{10}c_L$. This completes the proof. ∎

Following the Lemma C.1.2, we give the proof of Theorem 3.

**Proof of Theorem 3.** Denote $\delta_\lambda = \sqrt{\delta_n^2 + 2c_0^{-1}\lambda\|f^*\|_K^2}$ and note that $\mathcal{M}(\delta)/\delta$ is non-increasing in $\delta$, then

$$\frac{\mathcal{M}(\delta_\lambda)}{\delta_\lambda} \le \frac{\mathcal{M}(\delta_n)}{\delta_n} \le \frac{1}{2}c_0\delta_n \le \frac{1}{2}c_0\delta_\lambda,$$

where the second inequality follows from the definition of $\delta_n$. Then, we have $\mathcal{M}(\delta_\lambda) \le c_0\delta_\lambda^2/2$. In the following, we first establish the upper bound on $\mathcal{L}^2(P_{\mathbf{x}}^T)$-error by showing the following inequality holds conditioning on the event $\mathcal{K}(\delta_\lambda)$

$$\inf_{f\in\mathcal{H}_K, f\notin\Theta(\delta_\lambda)} \frac{1}{n}\sum_{i=1}^n \phi_n(\mathbf{x}_i)\left\{L(y_i, f(\mathbf{x}_i)) - L(y_i, f^*(\mathbf{x}_i))\right\} + \lambda\|f\|_K^2 - \lambda\|f^*\|_K^2 > 0, \quad (9)$$

where the definitions of $\mathcal{K}(\delta_\lambda)$ and $\Theta(\delta_\lambda)$ are provided in Lemma C.1.2. Note that it suffices to prove that (9) holds on the boundary of $\Theta(\delta_\lambda)$, denoted by $B(\Theta(\delta_\lambda))$. To see this, for any $f \in \mathcal{H}_K$ and $f \notin \Theta(\delta_\lambda)$, by the convexity of the two sets $\mathcal{H}_K$ and $\Theta(\delta_\lambda)$, there exists $0 < \alpha \le 1$ such that $\tilde{f} = \alpha f + (1-\alpha)f^* \in B(\Theta(\delta_\lambda))$. Applying Jensen's inequality yields

$$\phi_n(\mathbf{x}_i)\left\{L(y_i, \tilde{f}(\mathbf{x}_i)) - L(y_i, f^*(\mathbf{x}_i))\right\} + \lambda\|\tilde{f}\|_K^2 - \lambda\|f^*\|_K^2$$
$$\le \alpha\left\{\phi_n(\mathbf{x}_i)\left\{L(y_i, f(\mathbf{x}_i)) - L(y_i, f^*(\mathbf{x}_i))\right\} + \lambda\|f\|_K^2 - \lambda\|f^*\|_K^2\right\}.$$

Therefore, we only need to show

$$\frac{1}{n}\sum_{i=1}^n \phi_n(\mathbf{x}_i)\left\{L(y_i, f^*(\mathbf{x}_i))) - L(y_i, \tilde{f}(\mathbf{x}_i))\right\} + \lambda\|f^*\|_K^2 - \lambda\|\tilde{f}\|_K^2 < 0.$$

For $\tilde{f} \in B(\Theta(\delta_\lambda))$, we consider the following two cases: (i) If $\|\tilde{f} - f^*\|_T = \delta_\lambda$ and $\|\tilde{f} - f^*\|_K \le 3\|f^*\|_K$, we have

$$\frac{1}{n}\sum_{i=1}^n \phi_n(\mathbf{x}_i)\left\{L(y_i, f^*(\mathbf{x}_i))) - L(y_i, \tilde{f}(\mathbf{x}_i))\right\} + \lambda\|f^*\|_K^2 - \lambda\|\tilde{f}\|_K^2$$
$$\overset{(i)}{\le} \mathcal{M}(\delta_\lambda) - E_T\left[L(y, \tilde{f}(\mathbf{x})) - L(y, f^*(\mathbf{x}))\right] + \lambda\|f^*\|_K^2 - \lambda\|\tilde{f}\|_K^2$$
$$\overset{(ii)}{\le} \mathcal{M}(\delta_\lambda) - c_0\|\tilde{f} - f^*\|_T^2 + \lambda\|f^*\|_K^2 \le -\frac{c_0}{2}\delta_\lambda^2 + \lambda\|f^*\|_K^2 = -\frac{c_0}{2}\delta_n^2 < 0,$$

where the inequality (i) follows from Lemma C.1.2, (ii) is from Assumption 2. If $\|\tilde{f} - f^*\|_T \le \delta_\lambda$ and $\|\tilde{f} - f^*\|_K = 3\|f^*\|_K$, we have

$$\frac{1}{n}\sum_{i=1}^n \phi_n(\mathbf{x}_i)\left\{L(y_i, f^*(\mathbf{x}_i))) - L(y_i, \tilde{f}(\mathbf{x}_i))\right\} + \lambda\|f^*\|_K^2 - \lambda\|\tilde{f}\|_K^2$$
$$\le \mathcal{M}(\delta_\lambda) - E_T\left[L(y, \tilde{f}(\mathbf{x})) - L(y, f^*(\mathbf{x}))\right] + \lambda\|f^*\|_K^2 - \lambda\|\tilde{f}\|_K^2$$
$$\le \mathcal{M}(\delta_\lambda) + \lambda\|f^*\|_K^2 - \lambda\|\tilde{f}\|_K^2 \overset{(i)}{\le} \frac{c_0}{2}\delta_\lambda^2 - 3\lambda\|f^*\|_K^2 = \frac{c_0}{2}\delta_n^2 - 2\lambda\|f^*\|_K^2 \overset{(ii)}{<} 0,$$

where the inequality (i) follows from the fact that $\|\tilde{f}\|_K \ge 2\|f^*\|_K$ by triangle inequality, and the inequality (ii) follows from the definition of $\lambda$.

Combining the above results, the inequality (9) holds. Then, we can conclude that

$$\|\widehat{f}^\phi - f^*\|_T^2 \le \delta_\lambda^2 = \delta_n^2 + 2c_0^{-1}\lambda\|f^*\|_K^2$$

holds by the definition of $\widehat{f}^\phi$.

For the bound of excess risk, note that

$$
\begin{aligned}
&\mathcal{E}_T^L(\widehat{f}^\phi) - \mathcal{E}_T^L(f^*) \\
&\overset{(i)}{\le} \mathcal{E}_T^L(\widehat{f}^\phi) - \mathcal{E}_T^L(f^*) - \frac{1}{n}\sum_{i=1}^n \phi_n(\mathbf{x}_i)\big(L(y_i, \widehat{f}^\phi(\mathbf{x}_i)) - L(y_i, f^*(\mathbf{x}_i))\big) + \lambda\|f^*\|_K^2 - \lambda\|\widehat{f}^\phi\|_K^2 \\
&\overset{(ii)}{\le} \mathcal{M}(\delta_\lambda) + \lambda\|f^*\|_K^2 - \lambda\|\widehat{f}^\phi\|_K^2 = \mathcal{M}(\delta_\lambda) - 2\lambda\langle f^*, \widehat{f}^\phi - f^*\rangle_K - \lambda\|\widehat{f}^\phi - f^*\|_K^2 \\
&\le \frac{c_0\delta_\lambda^2}{2} + 2\lambda\|f^*\|_K\|\widehat{f}^\phi - f^*\|_K - \lambda\|\widehat{f}^\phi - f^*\|_K^2 \\
&\overset{(iii)}{\le} \frac{c_0\delta_\lambda^2}{2} + \lambda\|f^*\|_K^2 + \lambda\|\widehat{f}^\phi - f^*\|_K^2 - \lambda\|\widehat{f}^\phi - f^*\|_K^2 \\
&= \frac{c_0\delta_\lambda^2}{2} + \lambda\|f^*\|_K^2 = \frac{1}{2}c_0\delta_n^2 + 2\lambda\|f^*\|_K^2,
\end{aligned}
$$

where the inequality (i) follows from the definition of $\widehat{f}^\phi$, (ii) is from Lemma C.1.2, and (iii) is from the basic inequality. This completes the proof. ∎

## C.2 Proof of Theorem 2

To prove Theorem 2, we first provide a Lemma which is similar to Lemma C.1.2 and bounds the regular supremum of empirical process within a local ball.

**Lemma C.2.1.** *For any radii $\nu > 0$, define the event $\mathcal{K}'(\nu)$ as*

$$\sup_{f\in\Theta'(\nu)} \left| \frac{1}{n}\sum_{i=1}^n \left(L(y_i, f(\mathbf{x}_i)) - L(y_i, f^*(\mathbf{x}_i))\right) - E_S\left[L(y, f(\mathbf{x}) - L(y, f^*(\mathbf{x}))\right] \right| \le \mathcal{M}'(\nu),$$

*where $\Theta'(\nu) := \left\{ f \in \mathcal{H}_K \mid \|f - f^*\|_S \le c_0\nu^2/(c_L\sqrt{\beta^2}) \text{ and } \|f - f^*\|_K \le 3\|f^*\|_K \right\}$, then $\mathcal{K}'(\nu)$ holds with probability at least $1 - n^{-c_2}$.*

Note that

$$
\begin{aligned}
c_0\|f - f^*\|_T^2 &\overset{(i)}{\le} E_T\left[L(y, f(\mathbf{x})) - L(y, f^*(\mathbf{x}))\right] = E_S\left[\phi(\mathbf{x})\left(L(y, f(\mathbf{x})) - L(y, f^*(\mathbf{x}))\right)\right] \\
&\overset{(ii)}{\le} c_L\sqrt{\beta^2}\|f - f^*\|_S,
\end{aligned}
\tag{10}
$$

where the inequality (i) follows from Assumption 2, the inequality (ii) follows from Cauchy-Schwarz inequality and the fact that the loss function is $c_L$-Lipschitz continuous.

Denote

$$D = \sup_{f\in\Theta'(\nu)} \left| \frac{1}{n}\sum_{i=1}^n \left(L(y_i, f(\mathbf{x}_i)) - L(y_i, f^*(\mathbf{x}_i))\right) - E_S\left[L(y, f(\mathbf{x})) - L(y, f^*(\mathbf{x}))\right] \right|.$$

To bound $E[D]$, by following the similar argument as Lemma C.1.2, we have

$$E[D] \le \frac{4c_L}{n} E\left[ \sup_{f\in\Theta'(\nu)} \left| \sum_{i=1}^n \sigma_i(f(\mathbf{x}_i) - f^*(\mathbf{x}_i)) \right| \right].
\tag{11}$$

Denote $g = f - f^* \in \mathcal{H}_K$ and then, we have $g = \sum_{j=1}^{\infty} g_j \psi_j$ with $g_j = \int_{\mathcal{X}} f(\mathbf{x}) \psi_j(\mathbf{x}) \rho_{\mathbf{x}}^T(\mathbf{x}) d\mathbf{x}$ and

$$
\left| \sum_{i=1}^{n} \sigma_i (f(\mathbf{x}_i) - f^*(\mathbf{x}_i)) \right| = \left| \sum_{i=1}^{n} \sigma_i \sum_{j=1}^{\infty} g_j \psi_j(\mathbf{x}_i) \right|
$$

$$
= \left| \sum_{j=1}^{\infty} \frac{g_j}{\sqrt{\min(\nu^2, \mu_j \|f^*\|_K^2)}} \sqrt{\min(\nu^2, \mu_j \|f^*\|_K^2)} \sum_{i=1}^{n} \sigma_i \psi_j(\mathbf{x}_i) \right| \quad (12)
$$

$$
\overset{(i)}{\leq} \sqrt{10} \left\{ \sum_{j=1}^{\infty} \min(\nu^2, \mu_j \|f^*\|_K^2) \left( \sum_{i=1}^{n} \sigma_i \psi_j(\mathbf{x}_i) \right)^2 \right\}^{1/2},
$$

where the inequality (i) follows from Cauthy-Schwarz inequality and the fact that $\sum_{j=1}^{\infty} \frac{g_j^2}{\min(\nu^2, \mu_j \|f^*\|_K^2)} \leq 10$ by using $\|g\|_T \leq \nu$ from (10) and $\|g\|_K \leq 3\|f^*\|_K$.

Plugging (12) into (11), we have

$$
E[D] \leq \frac{4\sqrt{10}c_L}{n} E\Big[ \sum_{j=1}^{\infty} \min(\nu^2, \mu_j \|f^*\|_K^2) \left( \sum_{i=1}^{n} \sigma_i \psi_j(\mathbf{x}_i) \right)^2 \Big]^{1/2}
$$

$$
\overset{(i)}{\leq} \frac{4\sqrt{10}c_L}{n} \left\{ \sum_{j=1}^{\infty} \min(\nu^2, \mu_j \|f^*\|_K^2) E_{\mathbf{x},\sigma} \Big[ \sum_{i=1}^{n} \sigma_i \psi_j(\mathbf{x}_i) \Big]^2 \right\}^{1/2} \quad (13)
$$

$$
\overset{(ii)}{\leq} 4\sqrt{10}c_L R(\nu),
$$

where the inequality (i) follows from Jensen's inequality, the inequality (ii) is from the fact that $E_{\mathbf{x},\sigma}[\sigma_i \psi_j(\mathbf{x}_i)] = 0$, for each $i$ and the the assumption that $\|\psi_j\|_\infty \leq 1$, for all $j \geq 1$.

Next, we turn to bound the term $D - E[D]$ and following the similar argument as (12) yields that

$$
|g(\mathbf{x})| \leq \sqrt{\sum_{j=1}^{\infty} \frac{g_j^2}{\min(\nu^2, \mu_j \|f^*\|_K^2)}} \sqrt{\sum_{j=1}^{\infty} \min(\nu^2, \mu_j \|f^*\|_K^2) \psi_j^2(\mathbf{x})} \leq \sqrt{10n} R(\nu).
$$

Consequently, we have

$$
|L(y_i, f(\mathbf{x}_i)) - L(y_i, f^*(\mathbf{x}_i))| \leq c_L |g(\mathbf{x}_i)| \leq \sqrt{10n} c_L R(\nu),
$$

and

$$
E\left[(L(y_i, f(\mathbf{x}_i)) - L(y_i, f^*(\mathbf{x}_i)))\right]^2 \leq c_L^2 E\left[g^2(\mathbf{x}_i)\right] \leq 10n c_L^2 R^2(\nu).
$$

Then conditions in Lemma C.1.1 are satisfied with $\eta_n = \sqrt{10n} c_L R(\nu)$ and $\zeta_n^2 = 10n c_L^2 R^2(\nu)$. Let $t = \sqrt{\frac{c_2 \log n}{n}}$, with probability at least $1 - n^{-c_2}$, there holds that

$$
D - E[D] \leq \sqrt{\frac{c_2 \log n}{n} \left( 20n c_L^2 R^2(\nu) + 4c_L \sqrt{10n} R(\nu) E[D] \right)} + \frac{2\sqrt{10}c_2 c_L}{3} \frac{\log n}{\sqrt{n}} R(\nu)
$$

$$
\overset{(i)}{\leq} c_L \sqrt{\frac{20c_2 \log n}{n} (n + 8\sqrt{n})} R(\nu) + \frac{2\sqrt{10}c_2 c_L}{3} \frac{\log n}{\sqrt{n}} R(\nu) \quad (14)
$$

$$
\overset{(ii)}{\leq} \left( 3c_L \sqrt{20c_2} + \frac{\sqrt{20}c_2 c_L}{3} \right) \sqrt{\log n} R(\nu)
$$

where the inequality (i) follows from (13), and the inequality (ii) follows from the fact that $\frac{\log n}{n} < 1/2$, for $n \geq 2$.

Combining (13) and (14), with probability at least $1 - n^{-c_2}$, we have

$$
D \leq C \sqrt{\log n} R(\nu), \quad (15)
$$

where $C = 4\sqrt{10}c_L/\sqrt{\log 2} + 3c_L\sqrt{20c_2} + \sqrt{20}c_2c_L/3$. Thus we complete the proof by taking $\mathcal{M}'(\nu) = C\sqrt{\log n}R(\nu)$. ∎

**Proof of Theorem 2.** Let $\delta = c_0\nu^2/(c_L\sqrt{\beta^2})$, $\mathcal{Q}(\delta) = \mathcal{M}'\left((c_0^{-1}c_L\sqrt{\beta^2}\delta)^{1/2}\right)$, the function class $\mathcal{G}(\delta) = \Theta'\left((c_0^{-1}c_L\sqrt{\beta^2}\delta)^{1/2}\right)$ and the event $\mathcal{P}(\delta) = \mathcal{K}'\left((c_0^{-1}c_L\sqrt{\beta^2}\delta)^{1/2}\right)$. Denote $\delta_\lambda = \sqrt{\delta_n^2 + 2c_0^{-1}\lambda\|f^*\|_K^2}$ with $\mathcal{Q}(\delta) \le c_0\delta^2/2$. Since $\mathcal{M}'(\nu)/\nu$ is non-increasing in $\nu$, then it is easy to check $\mathcal{Q}(\delta)/\delta$ is non-increasing in $\delta$ by

$$\frac{\mathcal{Q}(\delta)}{\delta} = \frac{\mathcal{M}'\left((c_0^{-1}c_L\sqrt{\beta^2}\delta)^{1/2}\right)}{(c_0^{-1}c_L\sqrt{\beta^2}\delta)^{1/2}}\left\{c_0\delta/(c_L\sqrt{\beta^2})\right\}^{-1/2},$$

hence we also have $\mathcal{Q}(\delta_\lambda) \le c_0\delta_\lambda^2/2$. Following a similar treatment as that in the proof of Theorem 3, we can show that

$$\inf_{f \in \mathcal{H}_K, f \notin \mathcal{G}(\delta_\lambda)} \frac{1}{n}\sum_{i=1}^{n}\{L(y_i, f(\mathbf{x}_i)) - L(y_i, f^*(\mathbf{x}_i))\} + \lambda\|f\|_K^2 - \lambda\|f^*\|_K^2 > 0. \tag{16}$$

It implies by the definition of $\widehat{f}$ that $\|\widehat{f} - f^*\|_S \le \delta_\lambda$ with probability at least $1 - n^{-c_2}$. By (10), we have

$$\|\widehat{f} - f^*\|_T^2 \le c_0^{-1}c_L\sqrt{\beta^2}\delta_\lambda = c_0^{-1}c_L\sqrt{\beta^2}\sqrt{\delta_n^2 + 2c_0^{-1}\lambda\|f^*\|_K^2},$$

and

$$\mathcal{E}_T^L(\widehat{f}) - \mathcal{E}_T^L(f^*) = E_T\left[L(y, f(\mathbf{x})) - L(y, f^*(\mathbf{x}))\right] \le c_L\sqrt{\beta^2}\sqrt{\delta_n^2 + 2c_0^{-1}\lambda\|f^*\|_K^2}$$

with probability at least $1 - n^{-c_2}$. ∎

## C.3 Proof of Theorem 1

Note that the density ratio is bounded that $\sup_{\mathbf{x}\in\mathcal{X}}\phi(\mathbf{x}) \le \alpha$, which implies

$$\|f - f^*\|_T \le \sqrt{\alpha}\|f - f^*\|_S. \tag{17}$$

We can establish the similar result as that in Lemma (C.1.2) by taking $\Theta'(\nu) := \{f \in \mathcal{H}_K \mid \|f - f^*\|_S \le \nu/\sqrt{\alpha} \text{ and } \|f - f^*\|_K \le 3\|f^*\|_K\}$. Then, by choosing $\delta = \nu/\sqrt{\alpha}$ and following the similar treatment as that in the proof of Theorem 2, we have

$$\|\widehat{f} - f^*\|_S^2 \le \delta_n^2 + 2c_0^{-1}\lambda\|f^*\|_K^2.$$

with probability at least $1 - n^{-c_1}$, where $\delta_n$ satisfies $\mathcal{M}'(\sqrt{\alpha}\delta) = C\sqrt{\log n}R(\sqrt{\alpha}\delta) \le \frac{c_0\delta^2}{2}$. Together with (17), we have

$$\|\widehat{f} - f^*\|_T^2 \le \alpha\left(\delta_n^2 + 2c_0^{-1}\lambda\|f^*\|_K^2\right)$$

with probability at least $1 - n^{-c_1}$. On the other hand, by (10), there holds

$$\mathcal{E}_T^L(\widehat{f}) - \mathcal{E}_T^L(f^*) = E_T\left[L(y, f(\mathbf{x})) - L(y, f^*(\mathbf{x}))\right] \le c_L\alpha\sqrt{\left(\delta_n^2 + 2c_0^{-1}\lambda\|f^*\|_K^2\right)}$$

with probability at least $1 - n^{-c_1}$. Thus we complete the proof. ∎

**Remark C.3.1.** *One can combine the proofs of Theorem 1 and Theorem 2 to find out why the unweighted estimator for the bounded case achieves the optimal rate in terms of the $\mathcal{L}^2(P_\mathbf{x}^T)$-error, rather than only attaining sub-optimal for the moment bounded case. Both the two proofs first bound the supremum of empirical process under the classic regime, that is, without covariate shift. Then the fundamental distinction lies in that the inequalities (10) and (17) give two different convergence rates compared to $\|\widehat{f} - f^*\|_S$.*

## C.4 Proof of corollaries

**Proof of Corollary 1.** From the definition that $d(\delta) = \min\{j \geq 1 | \mu_j \leq \delta^2\}$ and the assumption that $\|f^*\|_K^2 = 1$, we have

$$\sum_{j=1}^{\infty} \min(\delta^2, \mu_j \|f^*\|_K^2) = \sum_{j=1}^{d(\delta)} \min(\delta^2, \mu_j) + \sum_{j=d(\delta)+1}^{\infty} \min(\delta^2, \mu_j) \leq d(\delta)\delta^2 + Cd(\delta)\delta^2 \asymp d(\delta)\delta^2,$$

where we use the definition of the regular kernel. So the inequality $C\sqrt{\log n}R(\sqrt{\alpha}\delta) \leq \frac{c_0 \delta^2}{2}$ can be simplified to

$$\sqrt{\frac{\alpha \log n}{n}} d(\sqrt{\alpha}\delta) \leq C\delta.$$

This proves the inequality (7) in the main text. For the finite-rank $D$ case, $\sum_{j=1}^{\infty} \min(\delta^2, \mu_j) \leq D\delta^2$, which implies

$$\delta_n^2 \leq C\frac{D\alpha \log n}{n}.$$

Combining the choice of $\lambda$ and Theorem 1 gives

$$\|\widehat{f} - f^*\|_T^2 \asymp \mathcal{E}_T^L(\widehat{f}) - \mathcal{E}_T^L(f^*) \leq C\frac{D\alpha^2 \log n}{n}.$$

For the eigenvalues with polynomial decay, such that $\mu_j \leq Cj^{-2r}$, so we have $d(\delta) \leq C(1/\delta)^{1/r}$, which implies that $\delta_n$ satisfies

$$\left(\frac{\log n}{n} \alpha^{\frac{2r-1}{2r}}\right)^{\frac{2r}{2r+1}} \leq C\delta^2.$$

The simple derivation leads to the desired result. Thus we complete the proof. ∎

**Proof of Corollary 2.** From the definition that $d(\delta) = \min\{j \geq 1 | \mu_j \leq \delta^2\}$ and the assumption that $\|f^*\|_K^2 = 1$, we have

$$\sum_{j=1}^{\infty} \min(\delta^2, \mu_j) = \sum_{j=1}^{d(\delta)} \min(\delta^2, \mu_j) + \sum_{j=d(\delta)+1}^{\infty} \min(\delta^2, \mu_j) \leq d(\delta)\delta^2 + Cd(\delta)\delta^2,$$

where we use the definition of the regular kernel. So the inequality $\mathcal{M}(\delta) \leq \delta^2/2$ can be simplified to

$$\sqrt{\frac{\beta^2 \log^2 n}{n}} d(\delta) \leq C\delta.$$

This also proves the inequality (16) in the main text. For the finite-rank $D$ case, $\sum_{j=1}^{\infty} \min(\delta^2, \mu_j) \leq D\delta^2$, which implies

$$\delta_n^2 \leq C\frac{D\beta^2 \log^2 n}{n}. \tag{18}$$

Combine (18), the choice of $\lambda$ and Theorem 3, we have

$$\|\widehat{f}^\phi - f^*\|_T^2 \asymp \mathcal{E}_T^L(\widehat{f}^\phi) - \mathcal{E}_T^L(f^*) \leq C\frac{D\beta^2 \log^2 n}{n}.$$

For the eigenvalues with polynomial decay, such that $\mu_j \leq Cj^{-2r}$, so we have $d(\delta) \leq C(1/\delta)^{1/r}$. According to inequality (16) in the main text, we have

$$\delta^2 + \frac{\beta^2 \log^2 n}{n} d(\delta) \leq \delta^2 + C\frac{\beta^2 \log^2 n}{n}(1/\delta)^{1/r},$$

which leads to an optimal choice $\delta^2 = C(\frac{\beta^2 \log^2 n}{n})^{\frac{2r}{2r+1}}$. ∎

**Proof of the sub-optimality of $\widehat{f}$ in moment bounded case.** Now, we come to verify the result in Table 1 for the unweighted estimator in the moment bounded case. For the kernel with finite $R$, the inequality $C\sqrt{\log n}R((c_L\sqrt{\beta^2}\delta/c_0)^{1/2}) \leq \frac{c_0\delta^2}{2}$ can be simplified to

$$\sqrt{\frac{\log n}{n}}D\sqrt{\beta^2}\delta \leq C\delta^2.$$

Simple derivation yields $\delta_n \leq (\frac{\log n}{n}D\sqrt{\beta^2})^{1/3}$. The desired convergence rate follows by setting $\lambda \asymp (\frac{D\sqrt{\beta^2}\log n}{n})^{2/3}$. For the eigenvalues with polynomial decay, by the argument as before, the inequality $C\sqrt{\log n}R((c_L\sqrt{\beta^2}\delta/c_0)^{1/2}) \leq \frac{c_0\delta^2}{2}$ can be simplified to

$$\sqrt{\frac{\log n}{n}}(\sqrt{\beta^2}\delta)^{\frac{2r-1}{2r}} \leq C\delta^2,$$

which leads to $\delta_n^2 \leq C(\frac{\log n}{n}(\beta^2)^{\frac{2r-1}{4r}})^{\frac{4r}{6r+1}}$. Thus we complete the proof by applying Theorem 2 with $\lambda \asymp (\frac{\log n}{n}(\beta^2)^{\frac{2r-1}{4r}})^{\frac{4r}{6r+1}}$. ∎

**Proof of the convergence rates with Gaussian kernel.** At last, for Gaussian kernel, the eigenvalues $\mu_j$ decay exponentially, that is $\mu_j \asymp e^{-Cj\log j}$ Bach & Jordan (2002). By the definition of $d(\delta)$, we have $d(\delta) \leq -C\log\delta^2$ for $0 < \delta < 1$. We first consider the moment bounded case. For the TIRW estimator, by applying $\sum_{j=1}^{\infty}\min(\delta^2, \mu_j) \leq Cd(\delta)\delta^2$, the inequality $C\sqrt{\beta^2\log n}R(\delta) \leq \frac{c_0\delta^2}{2}$ can be simplified to

$$C\beta^2\frac{(\log n)^2}{n}\log(1/\delta^2) \leq \delta^2,$$

which yields $\delta_n^2 \leq C\beta^2\frac{(\log n)^3}{n}$. With $\lambda \asymp \beta^2\frac{(\log n)^3}{n}$, Theorem 3 implies

$$\|\widehat{f}^\phi - f^*\|_T^2 \asymp \mathcal{E}_T^L(\widehat{f}) - \mathcal{E}_T^L(f^*) \leq C\beta^2\frac{(\log n)^3}{n}.$$

For the unweighted estimator, $C\sqrt{\log n}R((c_L\sqrt{\beta^2}\delta/c_0)^{1/2}) \leq \frac{c_0\delta^2}{2}$ can be simplified to

$$C\sqrt{\frac{\log n}{n}\log\left(\frac{1}{\sqrt{\beta^2}\delta}\right)}\sqrt{\beta^2}\delta \leq \delta^2,$$

which yields $\delta_n \leq C(\sqrt{\beta^2}\frac{(\log n)^2}{n})^{1/3}$. With $\lambda \asymp (\sqrt{\beta^2}\frac{(\log n)^2}{n})^{2/3}$, Theorem 2 implies

$$\|\widehat{f} - f^*\|_T^2 \asymp \mathcal{E}_T^L(\widehat{f}) - \mathcal{E}_T^L(f^*) \leq C(\beta^4\frac{(\log n)^2}{n})^{1/3}.$$

We next consider the uniformly bounded case. For the TIRW estimator, it is straightforward to obtain that

$$\|\widehat{f}^\phi - f^*\|_T^2 \asymp \mathcal{E}_T^L(\widehat{f}) - \mathcal{E}_T^L(f^*) \leq C\alpha\frac{(\log n)^3}{n}.$$

For the unweighted estimator, the inequality $C\sqrt{\log n}R(\sqrt{\alpha}\delta) \leq \frac{c_0\delta^2}{2}$ can be simplified to

$$C\sqrt{\frac{\log n}{n}\log\left(\frac{1}{\alpha\delta^2}\right)}\alpha\delta^2 \leq \delta^2,$$

which yields $\delta_n^2 \leq C\alpha\frac{(\log n)^2}{n}$. With $\lambda \asymp C\alpha\frac{(\log n)^2}{n}$, Theorem 1 implies

$$\|\widehat{f} - f^*\|_T^2 \asymp \mathcal{E}_T^L(\widehat{f}) - \mathcal{E}_T^L(f^*) \leq C\alpha^2\frac{(\log n)^2}{n}.$$

## C.5 Discussion on the minimax lower bound

Based on a standard application of Fano's inequality, Ma et al. (2023) establish a minimax lower bound for the regular kernel class by using the squared loss. For completeness of our paper, we present the relevant result below, which gives a relatively conservative lower bound by only taking the regression-based problems into consideration, which covers the squared loss and check loss. To be specific, we suppose that the conditional density of $\varepsilon := y - f^*(\mathbf{x})$ given $\mathbf{x}$ follows the normal distribution with mean zero and variance $\sigma^2$.

**Theorem C.5.1.** *For any $\alpha > 0$, there exists a pair of marginal distributions $(P_{\mathbf{x}}^S, P_{\mathbf{x}}^T)$ with $\alpha$-uniformly bounded importance ratio and an orthonormal basis $\{\psi_j\}_{j \geq 1}$ of $\mathcal{L}^2(\mathcal{X}, P_{\mathbf{x}}^T)$ such that for any regular kernel class with eigenvalues $\{\mu_j\}_{j \geq 1}$, we have*

$$\inf_{\widehat{f}} \sup_{f^* \in \mathcal{B}_{\mathcal{H}}(1)} E\left[\|\widehat{f} - f^*\|_T^2\right] \geq C \inf_{\delta > 0} \left\{\delta^2 + \sigma^2 \alpha \frac{d(\delta)}{n}\right\}, \tag{19}$$

*where $\mathcal{B}_{\mathcal{H}}(1) = \{f \in \mathcal{H}_K \mid \|f\|_K \leq 1\}$ represents the unit Hilbert ball.*

By simply comparing the lower bound in (19) to the upper bound in (7) and (14) in the main text, we can see that this lower bound is sharp since it is achieved by both the unweighted estimator and the TIRW estimator up to a logarithmic factor. And hence, in the uniformly bounded case, the unweighted estimator achieves minimax optimality, which indicates the TIRW estimator may not be necessary. In the moment bounded case, the upper bound in Theorem 3 also attains the lower bound in (19) up to logarithmic factors. For the reason that the second moment bounded class contains the uniformly bounded class, we can conclude the TIRW estimator is still preserving minimax optimality, whereas the unweighted estimator is far from optimal compared to the minimax lower bound.

## C.6 Remark about the importance ratio

It is worthy pointing out that in practice, it is unrealistic to obtain the true importance ratio $\phi(\mathbf{x})$, and it should be estimated from data, where we denote the estimator of $\phi(\mathbf{x})$ by $\widehat{\phi}(\mathbf{x})$. As illustrated in Section 4 of the main text, we adopted the KLIEP algorithm (Sugiyama et al., 2007b) to obtain $\widehat{\phi}(\mathbf{x})$ in all the numerical examples. While the theoretical results are established under the case that $\phi(\mathbf{x})$ is known. We want to emphasize that to the best of our knowledge, such a gap commonly appears in the existing literature, possibly due to inherent theoretical challenges. We decide to leave such a promising topic as potential future work, and we add some detailed discussions on the possible route for establishing the theoretical results. Specifically, the key step is that we need to bound the term

$$\sup_{f \in \Theta(\delta)} \left|(1/n) \sum_{i=1}^{n} (\widehat{\phi}_n(\mathbf{x}_i) - \phi_n(\mathbf{x}_i))(L(y_i, f(\mathbf{x}_i)) - L(y_i, f^*(\mathbf{x}_i)))\right|.$$

Thus the strong convergence rate of $\widehat{\phi}_n - \phi_n$ is required. It's important to note that the components within this term are not independent, as the estimated importance ratio $\widehat{\phi}$ relies on the source data. To address these intricacies, advanced technical tools are essential. Once we successfully bound this term, we can establish results similar to those presented in Theorem 3 by leveraging existing proof techniques with slight modification.