# OpenReview forum: "Towards a Unified Analysis of Kernel-based Methods Under Covariate Shift"
_NeurIPS.cc/2023/Conference — NeurIPS 2023 poster_

### Official Review · Reviewer_qhGh · 2023-06-30

**Soundness:** 4 excellent
**Presentation:** 3 good
**Contribution:** 3 good
**Rating:** 7
**Confidence:** 3

**Summary:**

This manuscript presents convergence rates for kernel methods under covariate shift. Results fit quite a general framework, including common classification and regression losses. Two approaches are analyzed: (i) a usual M-estimator and (ii) an importance-sampling-like M-estimator. It is shown theoretically and empirically that the latter outperform the former.

**Strengths:**

The analysis presented in this paper provides interesting theoretical results regarding learning under covariate shift, which is a contemporary topic. The manuscript is well organized; it explains clearly the problem, state the results while discussing the hypotheses and, at the end, illustrates the theoretical findings by a numerical experiment.
I would like to stress that discussions regarding hypotheses are opportune and corollaries provide intelligible results.
The take-home message, stating that the importance-sampling-like estimator is better that the naive one, is interesting and confirms practitioners’ intuition.

**Weaknesses:**

Major remarks:
1) My main concern is about the novelty of the proofs: hypotheses (i) and (ii) look like straightforward tools to link expectations under the source distribution to the target distribution by linearity or Cauchy-Schwarz inequality. I had a very quick glance to the supplementary material and it confirmed this guess (although I admit that I may be wrong). I think that its important, in order to assess the contribution of the paper, that the authors explain the original derivations appearing in the proofs, with respect to techniques used for obtaining similar results without covariate shift (unfortunately, I have no reference in mind).
2) Another (minor) point is that Figure 1 does not seem to verify neither hypothesis (i) nor (ii) since $\phi(x)$ seems to explode when $x \to \infty$. If it is the case, it would be better to find another example (or at least to discuss this point). If it is not the case, it would be informative to explain it.

Some suggestions of improvement:
1)  $f^*$ is defined in Section 2.1, before the problem setting in Section 2.2. However, in practice, it corresponds to the optimal function under the target distribution, which is not clearly stated. I suggest to make it clear.
2) Although an informed reader understand definitions Line 113, it is not totally clear that expectations are conditioned by observed data. I suggest to had this information.
3) $D$ could be added after “Finite rank” in Table 1.
4) Line 264, it is not totally clear that “For the moment bounded case” correspond to Figure 3. I suggest to had it.

Typographical remarks:
1) Extra “the” Line 5.
2) “that” instead of “that is” Lines 126, 131 and 132.
3) In Theorems 1-3, $\delta_n$ should satisfy an inequality that involves $\delta$ instead of $\delta_n$.
4) There are $\phi(\textrm x)$ instead of $\phi(\textrm x)^2$ Lines 212 and 223.
5) Full points are missing in captions.

**Questions:**

1) What is $f_j$ Line 154?
2) What does $\psi_j \le Cj^{-2r}$ (Lines 201 and 235) mean, given that $\psi_j$ is a function? Is a norm missing?
3) Does the trends (evolution with respect to $n$) observed in Figure 2 (b)-(e) and Figure 3 (b)-(e) is of the order $\left( \frac{\log n}{n} \right)^q$ or $\left( \frac{\log^2 n}{n} \right)^q$ as exhibited in Table 1?

**Limitations:**

Limitations are not addressed.

---

> ### Author Rebuttal · Authors · 2023-08-09
>
> Thank you very much for your constructive comments and valuable suggestions! Our point-by-point responses to your comments are given below.
>
> **Major remark 1**
>
> Thanks a lot for your concern with the novelty of our proofs. We admit that our proofs use many classical empirical process techniques, such as the concentration inequality  (Lemma C.1.1.) and the symmetrization technique (Eq.(10) in Section C of supplementary material, which is also popularly used for establishing the theoretical results without covariate shift [1,2]. Yet, we want to emphasize that there exist some significant differences in proofs between the traditional regime and the case under covariate shift. Following your suggestion, we have highlighted the proof ideas and motivations under the covariate shift and compared the difference with the case without covariate shift in the revised version.  Some key points are summarized as follows:
>
> Firstly, under the classical case without covariate shift, we only need to control the supremum over a functional class of empirical process, which is defined as the empirical sum minus its expectation counterpart, and then, some classical empirical process techniques such as symmetrization technique and Talagrand’s concentration inequality can be directly adopted. Yet, under the case with covariate shift, we need to control the difference between the empirical sum with the samples driven from the source model and the expectation under the target model, as shown in Lemma C.1.2 of Appendix. Thus, we first need to decompose the total error in the proof of Lemma C.1.2. into two parts: the first part $D_1$ is the classic empirical process formulation and the second part $D_2$  is the residual which doesn't exist in the case without covariate shift.  For $D_1$, the main difference is that we have an extra importance ratio term and we need some hypotheses (uniformly bounded or moment bounded) on the ratio to bound it using the Cauchy-Schwarz inequality and Lipschitz-continuous assumption. For $D_2$, we use a technical truncation strategy to bound it. The truncation $\gamma_n$ plays a crucial role in balancing the empirical error and the approximation error. In fact, a fast diverging $\gamma_n$ may reduce the approximation error but compromise the empirical error. We use the optimal truncation $\gamma_n=\sqrt{n\beta^2}$ which ensures the empirical error and approximation error converge at the same order up to a $\log$ factor. This proof strategy is significantly different from that under the classical case without covariate shift.
>
> Secondly, to establish the convergence rates for unweighted estimators in both uniformly bounded and moment bounded cases,  we can't trivially construct a constraint set associated with the $\mathcal{L}^2$-norm $\||f-f^*\||_T$ under the target distribution, as the classical case without covariate shift. To tackle this problem, we notice the relation between $\||f-f^*\||_S$ and $\||f-f^*\||_T$ as shown in Eq. (10) in Appendix, which motivates us to construct the constraint set $\Theta^\prime(\nu)$ associated with $\||f-f^*\||_S$ as in Lemma C.2.1. Also note that the coefficients $g_j$'s of the expansion for $g=f-f^*$ in the proof of Lemma C.2.1 corresponds to orthonormal eigenfunctions of $\mathcal{L}^2(\mathcal{X},P_x^T)$ instead of the $\mathcal{L}^2(\mathcal{X},P_x^S)$.
> Based on Lemma C.2.1, together with the definitions of $\mathcal{Q}(\delta), \mathcal{G}(\delta)$ and $\mathcal{P}(\delta)$ as illustrated in the proofs of Theorem 1 and 2, we successfully establish the convergence rates for unweighted estimators under covariate shift. This proof strategy is significantly different from that under the classical case without covariate shift.
>
> **Major remark 2**
>
> Thanks a lot for your suggestions. In the original version, Figure 1 serves as an illustration of the phenomenon of covariate shift, and indeed, it may not be a good example. It has been replaced by a new example, which satisfies hypothesis (ii), and more explanations have also been added in the revised version. The modified Figure 1 is also attached in the Author Rebuttal pdf.
>
> **Suggestions of improvement**
>
> Thank you for all your suggestions. 1. The reason for defining $f^*$ in Section 2.1 is to introduce the considered loss family. In the revised version, we have clearly stated that the true target function $f^*$ is defined under the target distribution in Section 2.2. 2. We have made it clear in the revised version by clarifying that the expectations are conditioned by the observed data. 3. ``Finite rank D' has been added in 'Table 1. 4. We will rewrite line 264 to “For the moment bounded case as demonstrated in (a) and (d) of Figure 3”.
>
> **Question 1 and 2**
>
> Thanks a lot for pointing out the notation mistakes. Actually, $f_j$ should be replaced by $a_j$ where $a_j$ is defined in Line 153,  $\psi_j$ should be replaced by $\mu_j$ where $\{\mu_j\}$'s are the eigenvalues of the kernel function defined in Line 151, and they have been corrected in the revised version.
>
> **Question 3**
>
> Thanks a lot for your comment. In the original version, although the Gaussian kernel is used in the simulation, we do not provide the theoretical result for the kernel class with exponential decay.  In the revised version, we have added the explicit convergence rates corresponding to this kernel class in Table 1 which is also attached in the Author Rebuttal pdf. It is thus clear from the table that the trends in Figure 2 (b)-(e) and Figure 3 (b)-(e) almost agree with our theoretical findings where the Gaussian kernel with exponential decay is used. Additional numerical results presented in the supplementary material also confirm the theoretical convergence rates.
>
> **Reference**
>
> [1] Yang, Y., Pilanci, M., & Wainwright, M. J. (2017). Randomized sketches for kernels: Fast and
> optimal nonparametric regression.
>
> [2] Lian, H. (2022). Distributed learning of conditional quantiles in the reproducing kernel Hilbert space.

---

> > ### Comment · Reviewer_qhGh · 2023-08-13
> >
> > I would like to thank the authors for their rebuttal, which I read carefully. The authors answered persuasively to my concern and provided a new introductory example, which satisfies Hypothesis (ii). I consequently agree to increase my score.

---

> > > ### Author Response · Authors · 2023-08-13
> > > **Thank you**
> > >
> > > Thank you very much for your reply and increasing the score! We appreciate your time and effort in reviewing our work.

---

### Official Review · Reviewer_ZoXA · 2023-07-05

**Soundness:** 3 good
**Presentation:** 4 excellent
**Contribution:** 3 good
**Rating:** 7
**Confidence:** 4

**Summary:**

The paper provides a unified analysis of convergence properties for different kernel-based estimators under covariate shift. The analysis covers different loss functions and is focused on standard and importance weighted empirical risk estimators. The former are specified in Eq. (1) and the latter in Eq. (3).

The first assumption is pretty standard and requires a uniformly bounded kernel function. The second assumption enforces a locally strong convexity on the expected loss function relative to source and target marginal distributions (source available during training, target assumed to be shifted and available at test time). The assumptions that characterize the distribution shift are given on page 4 (lines 131 and 132): i) in the first case the importance weights are $\alpha$-uniformly bounded, ii) in the second case the second moment of the importance weight function is bounded.

Theorem 1 gives convergence bounds relative to case i) under the assumptions above. Further assumption is made to give a more readable interpretation of the bound in Corollary 1 which ties the convergence rate to kernel spectrum decay.
Theorem 2 gives a similar convergence result in a more difficult case ii), again under the assumptions listed above.
Theorem 3 considers an estimator that uses importance weighted empirical risk estimator, with truncated importance weights to avoid issues with tail samples. It is for case ii) and bounded second moment of importance weights. The latter result indicates much tighter convergence rate than the one in Theorem 2 that considers standard estimator without importance weighting.
Empirical analysis illustrates the tightness of the bounds on synthetically generated learning tasks and a real-case study.

**Strengths:**

While I have not checked the proofs, the theoretical part of the paper is its strongest point. It is also an interesting characterization of distribution shift carried into the bounds and would be interesting to see what other more granular specifications are possible for future studies. A relative comparison between Theorem 2 and 3 also illustrates the utility of truncated importance weighted estimator, which might be important for practical applications.

**Weaknesses:**

Empirical study might be the weakest part of the paper but given its nature should be fine. It might also be interesting to see how relevant are the assumptions on distribution shift relative to practical applications and datasets.

**Questions:**

The formulation of theorems should be cleaned as currently there are symbols that have not been introduced properly. For instance, it is unclear what $\delta_n$ refers to here and how it is related to $\delta$. The authors have spent a fair amount of space to illustrate the bounds and allow for readers to build some intuition. However, still the clarity could be a bit improved by moving from Appendix C.4 the part that transforms Eq. (5) to (7). At first, I had the impression that the right hand side just will not converge under the assumption on $c_0$ and $\lambda$.

**Limitations:**

Adequately addressed

---

> ### Author Rebuttal · Authors · 2023-08-09
>
> We greatly appreciate your dedicated time reviewing our paper and valuable insights! Our point-by-point responses to your comments are given below.
>
> **Weakness**
>
> Thanks a lot for your valuable suggestions. We want to point out that due to the space limit, only a small fraction of numerical experiments are reported in the main text. Actually, a lot of numerical experiments on synthetic data and real data are presented in Section A of the supplementary material, where many other loss functions and multi-dimensional cases are considered. Moreover, we applied the proposed method to a wide range of real datasets and the numerical performance is summarized in Table 1 in Section A of the supplementary material. To some extent, this paper provides a comprehensive study of the numerical performance of the kernel-based methods under various scenarios with covariate shift.
>
> We agree with you that it is very interesting and meaningful to test if there exists a distribution shift in practical applications and datasets and if the distribution shift satisfies the uniformly bounded or moment bounded assumptions. Unfortunately, the relevant approaches have remained lacking to our best knowledge.  We decide to leave such a promising topic as potential future work. And it is interesting to note that as shown in our real applications, the TIRW estimator always outperforms the unweighted estimator, and thus we suggest using the TIRW estimator to analyze the real-life dataset.
>
> **Question**
>
> Thank you very much for your suggestions.  In the revised version, we have rearranged the formulation of theorems and added some more descriptions of the introduced symbols and obtained results to provide more intuitive explanations of the established theoretical results. Some of the modifications are listed below:
>
> Firstly, in Theorem 1, $\delta_n$ is defined as the smallest positive value $\delta$ satisfying  $C\sqrt{\log n} R(\sqrt{\alpha}\delta)\le c_0\delta^2/2$. Similar modifications have been made in other established results, including Theorems 2 and 3. Moreover, some useful discussions on $\delta_n$ have also been added just behind Theorem 1.
>
> Secondly, we have moved some detailed explanations from Appendix C.4 to clarify  the transformation from
>  Eq.(5) to (7) in the main text.  Specifically,  the kernel complexity function $R(\delta)$  can be well approximated by $C\sqrt{(1/n)d(\delta)\delta^2}$ for the regular kernel class that includes many widely used kernels, such as the kernels with polynomial or exponential decay in their eigenvalues. Then, the link from $R(\delta)$ to $d(\delta)$  can be used to derive Eq. (7). Moreover, the explicit formulation of $\delta^2_n$ corresponding to specific kernels can be obtained by using the bound of $d(\delta)$. For example, $d(\delta)\lesssim \log(1/\delta^2)$ with $0<\delta<1$ for Gaussian kernel and $d(\delta)\lesssim \delta^2$ for the finite rank kernel.  Once $\delta^2_n$ is determined, the upper bounds can be immediately obtained by applying theorems with the choice of $\lambda\asymp \delta_n^2$, and then we have $\||\widehat{f}-f^*\||^2_T\lesssim \delta^2_n$ from Eq.(5).

---

### Official Review · Reviewer_u3FW · 2023-07-06

**Soundness:** 3 good
**Presentation:** 3 good
**Contribution:** 3 good
**Rating:** 6
**Confidence:** 1

**Summary:**

The authors study the covariate shift setting of nonparametric (kernel) methods
(Regularized Empirical Risk Minimization with optional importance weighing) with
an analysis which includes a wide array of losses and and two conditions on the
importance function. They establish sharp convergence which corroborate other
rates in literature. Additionally they provide experiments showing these rates
in practice.


**Strengths:**

- Quality: The authors extend the results to a wide array of losses and two types of covariate shift problems which is nice.
- Clarity: The paper is well-written and notation makes it easy to follow


**Weaknesses:**

- There are quite a bit of terms which are unknown in practice and they need to
  estimate the importance function which limits the practical impact of the
  method.


**Questions:**

Could the authors clarify the discrepancy between the theorems and practice in terms of the importance function $\phi$? You mention that you use an estimated function as plugin, do you expect it should be possible to prove results when using a plugin instead of the real thing?


**Limitations:**

Same as in Questions section. No need for societal impact limitation.

---

> ### Author Rebuttal · Authors · 2023-08-09
>
> Thank you for your positive feedback and helpful comments on our work! Our point-by-point responses to your comments are given below.
>
> **Weakness**
>
> Thanks a lot for your comments. We admit that the true importance ratio is unknown in practical applications, which needs to be estimated from the unlabeled test data. However, we want to emphasize that in literature, many existing methods [1,2,3] can be directly applied to accurately estimate the importance ratio. As illustrated in our extensive numerical experiments, the numerical performances of the implemented method with the true importance ratio and the estimated importance ratio are negligible in both synthetic and real-life examples. This implies that the importance ratio can be efficiently estimated, which may help the practical implementation of the proposed method.
>
> **Question**
>
> Thank you very much for your comments. We want to clarify that our theoretical results are established with the known true importance ratio. Yet, as you mentioned in the previous comment, the importance ratio is unknown in practical applications. We admit that there exist some gaps between the theory and practice, but we also want to emphasize that such gaps commonly exist in the literature on covariate shift as far as we know [4,5]. We agree that it is very interesting to establish some similar theoretical results by using the plugin estimator and to our best knowledge, such results are still unknown in the literature, possibly due to the existing theoretical obstacles.  We decide to leave such a promising topic as potential future work, but add some detailed discussions on the possible route for establishing the theoretical results at the end of the supplemental file in the revised version.  Specifically, the key step is that we need to well control the term $\sup_{f \in \Theta(\delta)}|(1/n)\sum_{i=1}^n(\widehat{\phi}_n({\mathbf{x}}_i)-{\phi}_n({\mathbf{x}}_i))(L(y_i, f({\mathbf{x}}_i))-L(y_i, f^*({\mathbf{x}}_i))) |$, and thus the strong convergence rate of $\widehat{\phi}_n-{\phi}_n$ is required. We want to point out that the summands contained in this term are not independent since the estimated importance ratio $\widehat{\phi}$ uses the source data, and some advanced technical tools are needed. Once this term is well bounded, we can establish the results similar to Theorem 3 by using the existing proof technique with slight modification.
>
> **Reference**
>
> [1] Huang, J., Gretton, A., Borgwardt, K., Schölkopf, B., & Smola, A. (2006). Correcting sample selection bias by unlabeled data.
>
> [2] Gretton, A., Smola, A., Huang, J., Schmittfull, M., Borgwardt, K., & Schölkopf, B. (2009). Covariate shift by kernel mean matching.
>
> [3] Sugiyama, M., Nakajima, S., Kashima, H., Buenau, P., & Kawanabe, M. (2007b). Direct importance estimation with model selection and its application to covariate shift adaptation.
>
> [4] Tibshirani, R. J., Foygel Barber, R., Candes, E., & Ramdas, A. (2019). Conformal prediction under covariate shift.
>
> [5] Ma, C., Pathak, R., & Wainwright, M. J. (2023). Optimally tackling covariate shift in RKHS-based
> nonparametric regression.

---

> > ### Comment · Reviewer_u3FW · 2023-08-11
> > **Thank you**
> >
> > I appreciate the response from the authors and especially the detailed comment on how to potentially derive bounds when using the empirical importance ratio. I will keep my score.

---

> > > ### Author Response · Authors · 2023-08-12
> > > **Thank you for the feedback**
> > >
> > > Thank you for your feedback and all your comments! We appreciate your time and effort in reviewing our work.

---

### Official Review · Reviewer_bSPk · 2023-07-16

**Soundness:** 3 good
**Presentation:** 3 good
**Contribution:** 3 good
**Rating:** 5
**Confidence:** 4

**Summary:**

This paper studies the generalization guarantees of non-parameteric methods in RKHS under covariate shift.
Compared to previous work (Ma et al. AOS2023), the authors extend their results from the squared loss to general Lipchitz loss functions.
The derived results show that

- under the uniformly bounded case for the importantce ratio, the unweighted estimator achieves the optimal learning rates in the $L2(d PT)$ space, where $PT$ is the target distribution.
- under the bounded second monment case, the above estimator is sub-optimal.
- under a truncated ratio, a sharp learning rate can be achieved.


**Strengths:**

- generalization analysis under covariate shift is derived from the squared loss to general loss functions
- Under the the uniformly bounded case and bounded second moment case for the importantce ratio, the results can recover the result under the squared loss
- the results are derived under the truncated case

**Weaknesses:**

- Extension from the squared loss to general Lipschitz loss functions is based on Assumption 2. More discussion on this assumption is required for specific loss functions. If the space is limited, the discussion can be deferred to the appendix.

- There are several parts unclear in the proof. For example,  in the proof of Lemma C.1.2, the notations $P_n$ and $P$ are undefined in Eq.(2), and more details are needed for the first inequality in Eq. (2).

- The proof idea and structure is almost the same as (Ma et al. 2023). For example, there is no significant difference between the proof of Theorem 3 and Lemma 2 in (Ma et al. 2023). This is because, under Assumption 2 and Eq. (10), the results under Lipschitz loss functions can be well controlled.




**Questions:**

- Why does $\phi_n(x_i) \leq \phi(x_i)$ hold in line 193?

---

> ### Author Rebuttal · Authors · 2023-08-09
>
> Thank you for your thorough review of our paper and the valuable feedback you provided. We have carefully considered your comments and have made significant efforts to address each of your concerns.
>
> **weakness 1**
>
> Thanks a lot for your precious suggestion. Detailed discussions on Assumption 2 for specific loss functions have been added at the very beginning of the Appendix in the revised version. Precisely, recall that Assumption 2 is a local $c_0$-strongly convexity condition on the expected loss function with respect to $\mathcal{L}^2(\mathcal{X},P^S_x)$ and $\mathcal{L}^2(\mathcal{X},P^T_x)$ at $f^*$. So verifying Assumption 2 is equivalent to verifying the local $c_0$ strongly convexity of the loss function. Here are some examples: for the squared loss $L(y,f(\mathbf{x}))=(y-f(\mathbf{x}))^2$, note that for any $y \in \mathcal{R}$, the function $z \rightarrow (y-z)^2$ is strongly convex with parameter $c_0=1$, so $f \rightarrow L(y,f(\mathbf{x}))$ satisfies the condition in Assumption 2 with $c_0=1$.  For the Huber loss ${L}(y, f(\mathbf{x}))=(y-f(\mathbf{x}))^2$, if $|y-f(\mathbf{x})| \leq \delta; \delta|y-f(\mathbf{x})|-\frac{1}{2} \delta^2$, otherwise, since this loss function is locally equivalent to the squared loss, so it is locally strongly convex under mild tail condition on $y-f^*(\mathbf{x})$. For the check loss ${L}(y, f(\mathbf{x}))=(y-f(\mathbf{x}))\left(\tau-I_{\{y\leq f(\mathbf{x})\}}\right)$, the local strong convexity holds if the conditional density of $y-f^*(\mathbf{x})$ given $\mathbf{x}$ is bounded away from $0$ uniformly [1]. For other loss functions such as logistic loss, hinge loss, and so on, mode discussions and verifications are provided in Appendix, and we also refer to pages 470-472 in [2] for the details.
>
> **Weakness 2**
>
> Thanks a lot for your valuable comments on the notation and detail of the poof.  We have proofread the technical proof and all the necessary definitions and details have been added in the revised version.  Specifically, for the proof of Lemma C.1.2,  we have provided the definitions that $P\varphi:=E_{S}[\varphi(\mathbf{x},y)]$  and $P_n \varphi:=(1/n)\sum_i \varphi(\mathbf{x}_i,y_i)$ for the measurable function $\varphi(\mathbf{x},y)$ with $\varphi(\mathbf{x},y)=\phi_n(\mathbf{x})(L(y,f(\mathbf{x}))-L(y,f^*(\mathbf{x})))$ in Eq.(2), and clearly clarified that the expectation $E[\cdot]$ is taking with respect to all random variables contained in it.  Moreover, detailed explanations for the first inequality in Eq. (2) have also been added and some reference has also been properly provided, such as Proposition 4.11 in [2]. Precisely, it is derived from the symmetrization technique, which states that for any class of measurable functions $\mathcal{F}$,  we have $E[\sup (P_n-P)\varphi]\le 2E[ \sup (1/n) \sum_i \sigma_i \varphi({\mathbf{x}}_i,y_i)  ] $, where $\sigma_i$'s are the Rademacher variables. Additionally, it is necessary to stress that the coefficients $g_j$'s of the expansion for $g=f-f^*$ in the proof of Lemma C.2.1 corresponds to orthonormal eigenfunctions of $\mathcal{L}^2(\mathcal{X},P_x^T)$ instead of $\mathcal{L}^2(\mathcal{X},P_x^S)$.
>
> **Weakness 3**
>
> Thanks a lot for your comments.  We admit that work in [3] makes tremendous contributions to the theories for the squared loss function under covariate shift and it is true that the Lipschitz property of loss functions and Assumption 2 play a crucial role in establishing our theoretical results. Yet, we want to emphasize that there still exist significant theoretical gaps from extending the squared loss function to the general loss function.
>
> It is well-known that the estimator of KRR with squared loss has an explicit solution. Then, [3] takes advantage of the analytic solution to establish several critical results, including the theoretical bounds similar to Theorem 1 and 2 in our paper. Yet,
> when the general loss function is considered, such an explicit solution does not exist, and different theoretical treatments are needed. Specifically, we turn to another proof strategy by noting the relation between $\||f-f^*\||_S$ and $\||f-f^*\||_T$ as shown in Eq. (10), which motivates us to establish Lemma C.2.1 where the constraint set $\Theta^\prime(\nu)$ is associated with the $\mathcal{L}^2$-norm $\||f-f^*\||_S$,  which is significantly different from Lemma C.1.2 and the traditional practice by using $\||f-f^*\||_T$. Based on Lemma C.2.1, together with the definitions of $\mathcal{Q}(\delta), \mathcal{G}(\delta)$ and $\mathcal{P}(\delta)$ as in the proofs of Theorem 1 and 2, we successfully establish the convergence rates for unweighted estimators under covariate shift. It is worth pointing out that the convergence rate established in [3] can be regarded as a special case of our theoretical result,  which matches the upper bounds in Theorem 1 and 2.  Additionally,  we also established the sharp bounds in terms of excess risk under various cases, which provides a  unified analysis of kernel-based methods under covariate shift.
>
> We want to emphasize that to some extent, this paper also provides a comprehensive study on the numerical performance of the kernel-based methods under various scenarios with covariate shift, which lends further support to the unified analysis and also presents the contributions of this paper.
>
> **Question**
>
> Thanks a lot for your comment. In fact, $\phi_n({\mathbf{x}}_i)$ is the truncation version of $\phi({\mathbf{x}}_i)$ that is $\phi_n(\mathbf{x})=\min\left(\phi(\mathbf{x}),\gamma_n\right)$ as defined in Line 142 of the main text, and thus the inequality $\phi_n({\mathbf{x}}_i)\leq \phi({\mathbf{x}}_i)$ holds directly.
>
> **Reference**
>
> [1] Lian, H. (2022). Distributed learning of conditional quantiles in the reproducing kernel Hilbert space.
>
> [2] Wainwright M J. High-dimensional statistics: A non-asymptotic viewpoint.
>
> [3] Ma, C., Pathak, R., & Wainwright, M. J. (2023). Optimally tackling covariate shift in RKHS-based
> nonparametric regression.

---

> > ### Comment · Reviewer_bSPk · 2023-08-12
> >
> > Thanks for the authors' response. It addressed most of my concerns and I increase my score to 5.

---

> > > ### Author Response · Authors · 2023-08-12
> > > **Thank you**
> > >
> > > Thank you very much for your feedback and increasing the score! We appreciate your time and effort in reviewing our work.

---

### Author Rebuttal · Authors · 2023-08-09

Thank you sincerely for your insightful comments and for dedicating your valuable time and effort toward the thorough evaluation of our paper.  We have carefully considered all questions, concerns, and comments raised by the reviewers. The insights and suggestions from reviewers have greatly contributed to enhancing the quality and clarity of our work. And we provided detailed responses to each review separately.  We also meticulously incorporated our responses into the revised paper and supplementary materials, mainly covering the following aspects: to highlight the contributions of this paper from the aspects of theory and practical application; to provide more details and insightful understanding about the established results; to correct all the typos, add some more descriptions of the definitions and make clear the introduced symbols. The attached pdf is a revised figure and table (Reviewer qhGh). Once again, we extend our heartfelt gratitude for your time, expertise, and contribution to our work.

---

### Decision · Program_Chairs · 2023-09-21

**Decision:**

Accept (poster)

**Comment:**

The reviewers unanimously recommended to accept the paper.